# A single cell transcriptional roadmap of human pacemaker cell differentiation

Alexandra Wiesinger[1], Jiuru Li[1], Lianne Fokkert[1], Priscilla Bakker[1], Arie O Verkerk[1,2], Vincent M Christoffels[1], Gerard JJ Boink[1,3], Harsha D Devalla[1]*

[1]Department of Medical Biology, Amsterdam University Medical Centers, University of Amsterdam, Amsterdam, Netherlands; [2]Department of Experimental Cardiology, Amsterdam University Medical Centers, University of Amsterdam, Amsterdam, Netherlands; [3]Department of Cardiology, Amsterdam University Medical Centers, University of Amsterdam, Amsterdam, Netherlands

**Abstract** Each heartbeat is triggered by the sinoatrial node (SAN), the primary pacemaker of the heart. Studies in animal models have revealed that pacemaker cells share a common progenitor with the (pro)epicardium, and that the pacemaker cardiomyocytes further diversify into 'transitional', 'tail', and 'head' subtypes. However, the underlying molecular mechanisms, especially of human pacemaker cell development, are poorly understood. Here, we performed single cell RNA sequencing (scRNA-seq) and trajectory inference on human induced pluripotent stem cells (hiPSCs) differentiating to SAN-like cardiomyocytes (SANCMs) to construct a roadmap of transcriptional changes and lineage decisions. In differentiated SANCM, we identified distinct clusters that closely resemble different subpopulations of the in vivo SAN. Moreover, the presence of a side population of proepicardial cells suggested their shared ontogeny with SANCM, as also reported in vivo. Our results demonstrate that the divergence of SANCM and proepicardial lineages is determined by WNT signaling. Furthermore, we uncovered roles for TGFβ and WNT signaling in the branching of transitional and head SANCM subtypes, respectively. These findings provide new insights into the molecular processes involved in human pacemaker cell differentiation, opening new avenues for complex disease modeling in vitro and inform approaches for cell therapy-based regeneration of the SAN.

*For correspondence:
h.d.devalla@amsterdamumc.nl

## Editor's evaluation

The manuscript by Wiesinger et al., demonstrates the differentiation of human induced pluripotent stem cells (iPSCs) into pacemaker cardiomyocytes. Authors have shown impressive analyses of sinoatrial node cardiomyocytes (SANCM) using scRNA-seq followed by a computational method namely Trajectory Inference (TI) to understand the diversification of SAN subtypes. The study further shows a key role for Wnt signaling in the critical branching of pacemaker cardiomyocytes and/or pro-epicardial cells. The authors also demonstrate that active TGFβ signaling promotes differentiation towards SAN transitional cells.

## Introduction

The human heart beats about 3 billion times in an average life span. Each heartbeat is triggered by the electrical impulses generated by the sinoatrial node (SAN), referred to as the primary pacemaker of the heart. Dysfunction of the SAN results in potentially life-threatening bradyarrhythmia (*Choudhury et al., 2015*) and current treatment with the implantation of electronic pacemakers is suboptimal (*Cingolani et al., 2018*; *Boink et al., 2015*). A better understanding of the origin, composition, and

function of the human SAN will enable the development of effective therapies. Previous studies have revealed that the SAN is a complex heterogeneous structure composed of both myocardial (head, tail, transitional cells) and non-myocardial cells such as fibroblasts, smooth muscle cells, etc., which contribute to its function (*Wiese et al., 2009*; *Bressan et al., 2018*; *Goodyer et al., 2019*). The mechanisms that regulate the development of the various cell types of the SAN niche remain largely unknown.

The pacemaker cells of the SAN originate from a *Tbx18*⁺ progenitor population that also gives rise to proepicardial cells (*van Wijk et al., 2009*; *Mommersteeg et al., 2010*). Moreover, proepicardium-derived mesenchymal cells have been shown to be integral for remodeling and sustained electrical activity of the SAN (*Bressan et al., 2018*). In chicken development, bone morphogenetic protein (BMP) and fibroblast growth factor (FGF) signaling have been shown to orchestrate the separation of myocardial and proepicardial cells (*Kruithof et al., 2006*; *van Wijk et al., 2009*). In vitro studies using human pluripotent stem cells point to a crosstalk between BMP, retinoic acid (RA) and wingless-related integration site (WNT) signaling in the differentiation of pacemaker and proepicardial cells (*Wiesinger et al., 2021*).

Within the cardiomyocyte fraction of the SAN, there are distinct subpopulations such as head, tail, and transitional cells (*Komosa et al., 2021*). The pacemaker cells in the SAN-head population express the T-box transcription factors *Tbx18* and *Tbx3*. This region is also distinct from all other cardiomyocytes in the heart as it lacks the expression of *Nkx2-5* (*Wiese et al., 2009*). The SAN-tail located inferior to the SAN-head expresses *Tbx3* and *Nkx2-5* but is devoid of *Tbx18* (*Wiese et al., 2009*; *Goodyer et al., 2019*). Furthermore, transitional cells (SAN-TZ) with transcriptional and functional properties intermediate to that of pacemaker cells and the adjacent atrial myocardium have also been reported, which are believed to play a critical role in transmitting the electrical impulses from the SAN to the adjacent atrial myocardium (*Boyett et al., 2000*; *Csepe et al., 2016*; *Goodyer et al., 2019*; *Li et al., 2019*).

While the shared origin of pacemaker cells with proepicardium and further differentiation of the pacemaker cells to distinct subpopulations is recognized, the mechanisms underlying these processes are poorly understood. Moreover, the vast majority of data regarding the development of the SAN is derived from animal models (*van Eif et al., 2018*) and our insights into human SAN development are very limited (*Csepe et al., 2016*; *Chandler et al., 2009*; *Sizarov et al., 2011*; *van Eif et al., 2019*). Differentiating human induced pluripotent stem cells (hiPSCs) are an excellent model to study human heart development in vitro providing easy access to early developmental stages and allowing the reconstruction of cell fate decisions.

Here, we show that the differentiation of hiPSCs to SAN cardiomyocytes (SANCM) recapitulates developmental programs with remarkable fidelity. Single cell RNA sequencing (scRNA-seq) demonstrated that the differentiated cell pool contains myocardial populations resembling pacemaker cell types in the different subdomains of the in vivo SAN, that is, SAN-head, SAN-tail, and SAN-TZ cells, in addition to a non-myocardial side population of proepicardial cells, reflecting their shared ontogeny. Using trajectory inference analysis tool URD, we provide a transcriptional roadmap of these cell types and identify that the fate decision of a common progenitor toward myocardial or proepicardial lineages is determined by WNT signaling. Importantly, our approach allowed the identification of signaling pathways involved in the divergence of SANCM subpopulations. Leveraging this data, we further show that active TGFβ signaling directs differentiation exclusively toward SAN-TZ cells.

Our results provide insight into the early specification and diversification of human pacemaker cells. The ability to obtain the various subpopulations and steer this differentiation process offers opportunities for assembling advanced in vitro models to better understand SAN function in health and disease and will further strengthen the basic framework for the development of regenerative therapies.

## Results

### Differentiation of hiPSCs to sinoatrial nodal and ventricular cardiomyocytes

Differentiation of hiPSCs toward *MESP1*⁺ mesoderm was initiated by activating Activin/Nodal, BMP and WNT signaling, as previously described (*Devalla et al., 2015*; *Devalla et al., 2016*). To steer

mesoderm toward a cardiomyocyte fate, WNT signaling was inhibited using XAV 939 for 96 hr, which resulted in predominantly ventricular-like cardiomyocytes (VCMs). To direct mesoderm toward SANCM, we treated cultures with BMP4, RA, WNT inhibitor (XAV 939), FGF inhibitor (PD173074), and ALK5 inhibitor (SB431542) for 48 hr from day 4 to day 6 (*Figure 1A*; *Protze et al., 2017*). Contracting cardiomyocytes were observed from day 10 onward and phenotypical differences in beating rates were apparent; SANCM monolayers exhibited faster beating rates in contrast to slower beating rates of VCM monolayers. TNNT2 expression was used as a measure of differentiation efficiency and flow cytometry analysis on day 19 demonstrated the presence of 60–90% cardiomyocytes in both groups (*Figure 1B*).

To assess cardiomyocyte identity, gene expression profiling was performed by RT-qPCR. Both cardiomyocyte subtypes expressed sarcomeric genes *TNNT2*, *ACTN2,* and the transcription factor *NKX2-5* (*Figure 1C*). Although *NKX2-5* expression was generally lower in SANCM compared with VCM, the difference was not statistically significant. The expression of transcription factors *SHOX2*, *TBX3*, *TBX18,* and *ISL1*, each required for proper SAN function (*van Eif et al., 2018*), was significantly higher in SANCM, indicating a SAN-like phenotype (*Figure 1D* and *Figure 1—figure supplement 1A*). VCM identity was verified by the expression of genes enriched in the ventricles, such as *MYL2*, *HOPX,* and *MYH7* (*Figure 1E*). In line with the above findings, immunofluorescence staining confirmed that SHOX2 and ISL1 are predominantly expressed in SANCM, whereas MYL2 expression was exclusively found in VCM (*Figure 1F, G* and *Figure 1—figure supplement 1B*).

## SANCM and VCM display distinct electrophysiological properties

Besides transcription factors, a number of ion channel genes are differentially expressed between the SAN and the ventricles, which confer distinct electrophysiological properties. The expression of *HCN1* and *HCN4*, which contribute to cardiac funny current $I_f$, implicated in pacemaking, was significantly higher in SANCM compared with VCM. Similarly, the L-type and T-type $Ca^{2+}$ channel genes *CACNA1D* and *CACNA1G*, respectively, as well as the inward rectifying $K^+$ channel Kir3.1, encoded by *KCNJ3*, were significantly upregulated in SANCM compared with VCM (*Figure 2A*). On the contrary, expression of *SCN5A*, the gene encoding cardiac $Na^+$ channel NaV1.5, was higher in VCM (*Figure 2A*). Consistently, action potential parameters (analyzed as in *Figure 2B*) of SANCM and VCM measured by single cell patch clamp confirmed expected subtype-specific electrophysiological differences. Representative traces of spontaneous action potentials are shown in *Figure 2C*, demonstrating shorter cycle length in SANCM (496.6±33.0 ms, mean ± s.e.m., N=12) compared with VCM (1241.5±111.7 ms, N=12) (*Figure 2D*). Consistent with a SAN phenotype, the maximum diastolic potential (MDP) was less negative in SANCM (–62.5±1.9 mV) compared with VCM (–69.9±1.4 mV). Furthermore, SANCM displayed a lower action potential amplitude (APA) and slower upstroke velocity (Vmax; 5.2±0.9 V/s SANCM versus 23.1±3.7 V/s VCM). Notably, MDPs and Vmax recorded in SANCM are similar to freshly isolated human SAN cells (*Verkerk et al., 2007*). On the contrary, longer action potential durations (APDs) at 20%, 50%, and 90% repolarization (APD20, APD50, and APD90, respectively) characterized the VCM (*Figure 2D* and *Figure 2—source data 1*). In addition, treatment with 3 µM ivabradine (IVA), an $I_f$ channel blocker (*Bucchi et al., 2002*), resulted in a significant increase in cycle length in SANC-Mtra (baseline [BL]: 491.1±76.8 ms; IVA: 771.9±124.7 ms, N=6), whereas cycle length in VCM was unaffected (BL: 838.0±110.9 ms; IVA: 817.9±114.0 ms, N=6) (*Figure 2E* and *Figure 2—source data 1*). Taken together, these results affirm the cellular identities expected for SANCM and VCM.

## Unmasking the cellular compositions in SANCM and VCM cultures

The variations in the expression of key genes, such as *TBX18* in SANCM and *MYL2* in VCM, are suggestive of heterogeneity in cellular composition (*Figure 1D and E*). In order to better understand the basis for this, we performed scRNA-seq according to the SORT-seq protocol (*Muraro et al., 2016*). A total of 1287 cells passed pre-processing and quality control. Since plate-to-plate variations were observed, the dataset was corrected using the standard integration workflow on SCTransform normalized data (*Figure 3—figure supplement 1*; *Hafemeister and Satija, 2019*; *Stuart et al., 2019*). Next, unsupervised clustering was performed with the top 15 principal components (PCs), which identified 12 clusters. One of the 12 clusters (cluster 9) showed enriched expression of spike-in DNA/ERCCs (*Figure 3—figure supplement 1B*), indicating the amplification of mostly ambient RNA and was therefore excluded from further analysis. We also removed two other small clusters (clusters

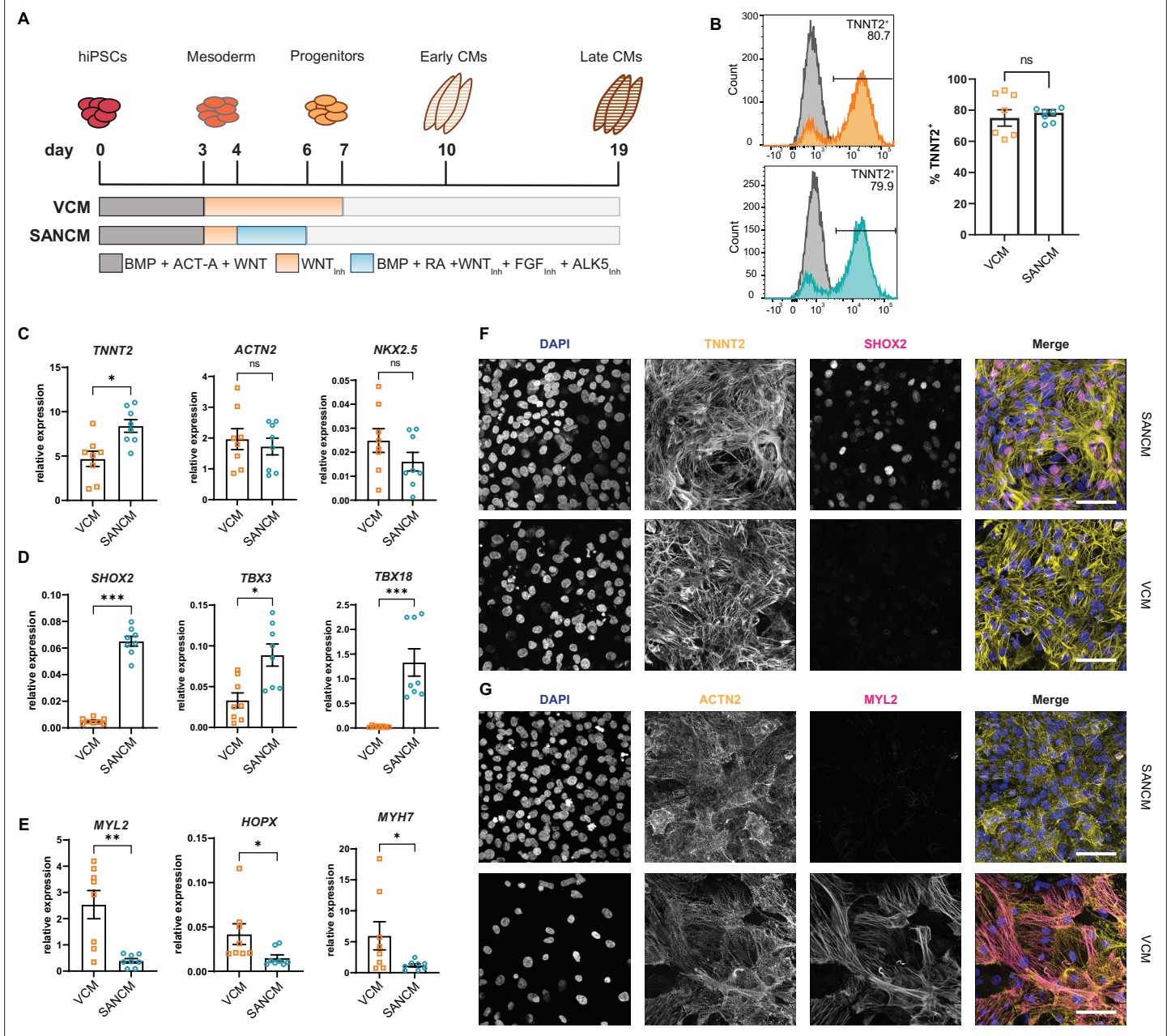

**Figure 1.** Differentiation of human induced pluripotent stem cells (hiPSCs) to sinoatrial node (SANCM) and ventricular-like cardiomyocytes (VCM). (**A**) Schematic representation of protocols used to differentiate hiPSCs to VCM and SANCM. (**B**) Representative histograms (left) and summarized data (right) showing percentage of TNNT2$^+$ cells in VCM (orange) and SANCM (blue) at day 19 of differentiation. A corresponding IgG isotype antibody was used as negative control for flow cytometry (gray). N=7 independent differentiations. Error bars, s.e.m. Mann-Whitney U test: p>0.05 (ns). (**C–E**) RT-qPCR depicting expression of pan cardiomyocyte genes (**C**), SAN-associated genes (**D**), and ventricular-associated genes (**E**) at day 19 of differentiation. N=8 independent differentiations; corrected to GEOMEAN of reference genes RPLP0 and GUSB. Error bars, s.e.m. Mann-Whitney U test: p<0.05 (*), p<0.005 (**), p<0.0005 (***). (**F–G**) Immunofluorescence stainings demonstrating the expression of nuclear stain DAPI, SHOX2, and TNNT2 (**F**), MYL2 and ACTN2 (**G**), in SANCM and VCM. Scale bars, 50 µm. Also see *Figure 1—figure supplement 1*.

The online version of this article includes the following figure supplement(s) for figure 1:

**Figure supplement 1.** Expression of ISL1 in sinoatrial node-like cardiomyocyte (SANCM) and ventricular-like cardiomyocyte (VCM).

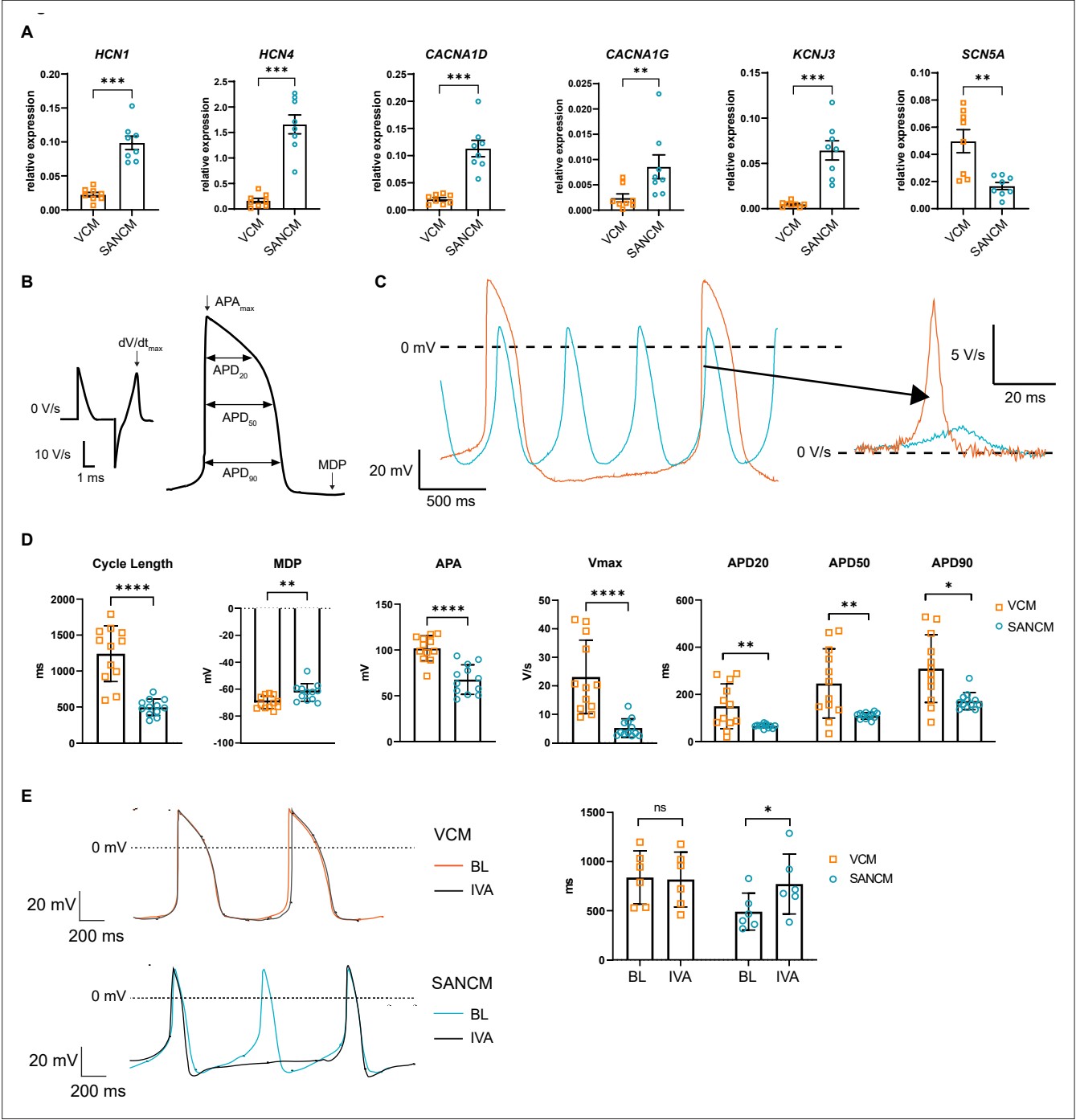

**Figure 2.** Electrophysiological characterization of sinoatrial node-like cardiomyocyte (SANCM) and ventricular-like cardiomyocyte (VCM). (**A**) RT-qPCR showing expression of ion channel genes at day 19 of differentiation. N=8 independent differentiations; corrected to GEOMEAN of reference genes RPLP0 and GUSB. Error bars, s.e.m. Mann-Whitney U test: p<0.05 (*), p<0.005 (**), p<0.0005 (***). (**B**) Action potential (AP) illustration depicting analyzed electrophysiological parameters. (**C**) Representative traces of spontaneous APs of day 19 SANCM (blue) and VCM (orange). (**D**) Cycle length, MDP, APA, Vmax, and APD20, APD50, and APD90 of VCM and SANCM at day 19 of differentiation. N=12 cells from four independent differentiations. Error bars, s.e.m. Mann-Whitney U test: p<0.05 (*), p<0.005 (**), p<0.0001 (****). (**E**) Cycle lengths of SANCM and VCM measured at baseline (BL) and after treatment with 3 μM ivabradine (IVA). N=6 cells from three independent differentiations. Error bars, s.e.m. Wilcoxon signed-rank test: p<0.05 (*). MDP, maximal diastolic potential; APA, action potential amplitude; Vmax, upstroke velocity; APD20, APD50, APD90, AP duration at 20%, 50%, 90% repolarization, respectively. Also see related source data file *Figure 2—source data 1*.

The online version of this article includes the following source data for figure 2:

**Source data 1.** Electrophysiological characterization of SANCM and VCM.

10 and 11), which showed enrichment in cell cycle-associated genes and genes associated with extra-ocular muscle development, respectively (*Figure 3—figure supplement 1B*). The remaining nine clusters (comprised of 1083 cells) were visualized using uniform manifold approximation and projection (UMAP) (*McInnes et al., 2018*; *Figure 3A*). The majority of the clusters highly expressed cardiac sarcomeric genes such as *TNNT2* and *ACTN2*, validating cardiomyocyte identity (*Figure 3B*). Clusters containing cells from the VCM differentiation protocol (clusters 0–3) did not overlap with cell clusters from the SANCM protocol (clusters 4–8), confirming the generation of transcriptionally different cardiomyocyte subtypes (*Figure 3—figure supplement 1A*). In addition, we observed that the non-cardiomyocyte side populations are specific for each differentiation protocol (*Figure 3A and B* and *Figure 3—figure supplement 1A*). List of genes differentially expressed in each cluster are provided in *Supplementary file 1*.

Analysis of cell clusters belonging to the VCM group unmasked the presence of three cardiomyocyte populations (clusters 1, 2, 3) and one non-cardiomyocyte population (cluster 0) as determined by the expression of sarcomeric genes *TNNT2* and *ACTN2* (*Figure 3B*). Clusters 2 and 3 expressed *MYH7* and *MYL2* indicating their ventricular identity (*Figure 3C* and *Figure 3—figure supplement 1C*). However, we observed differences in the expression of other ventricular genes between these two clusters. While the expression of *HOPX* was higher in cluster 2, *HEY2* and *IRX4* expression was restricted to cluster 3 (*Figure 3—figure supplement 1C*). The abundant expression of *HOPX* in cluster 2 likely may represent a more mature cardiomyocyte pool as reported in other similar studies (*Churko et al., 2018*; *Friedman et al., 2018*). Nevertheless, the top 10 differentially expressed genes of clusters 2 and 3 greatly overlap (*Figure 3C*) and differences in gene expression may also be the result of different transcriptional states resulting from transcriptional bursts.

Cluster 1 in the VCM group clusters closely with the ventricular cardiomyocytes (clusters 2 and 3). However, in contrast to the other clusters, it showed lower expression of typical cardiac genes (*Figure 3B*). Furthermore, cluster 1 was characterized by the expression of genes such as *HAPLN1*, *GPC3*, and *SEMA3C* (*Figure 3C* and *Figure 3—figure supplement 1D*), which are associated with progenitors of the myocardial embryonic outflow tract/right ventricle (*Sahara et al., 2019*; *Liu et al., 2019*). An embryonic outflow tract-like cellular identity of cluster 1 is further supported by the expression of *BMP4*, *ISL1*, and *PITX2* (*Figure 3D* and *Figure 3—figure supplement 1D*) and is consistent with previous findings describing co-differentiation of outflow tract-like cells with VCM (*Friedman et al., 2018*). Lastly, the non-cardiomyocyte cluster 0 expressed *NFATC1*, *FOXC1*, *NRG1*, and *NPR3* (*Figure 3—figure supplement 1E*), thus representing a fetal endocardial-like lineage (*Mikryukov et al., 2021*).

The SANCM population revealed four cardiomyocyte clusters (clusters 4–7), marked by *TNNT2* and *ACTN2* expression, and a smaller non-cardiomyocyte cluster (cluster 8) (*Figure 3A-C*). Cardiomyocyte clusters 4–6 expressed SAN-associated transcription factors, *TBX3* and *ISL1*, as well as *BMP4*, a SAN-enriched BMP signaling ligand (*van Eif et al., 2019*), albeit at varying levels (*Figure 3D*). However, *SHOX2* and *RGS6*, encoding a regulator of parasympathetic signaling in heart (*Goodyer et al., 2019*; *Yang et al., 2010*), were restricted to clusters 5 and 6 (*Figure 3D*). Moreover, we observed two salient differences between clusters 5 and 6. While cluster 6 expressed *TBX18* besides other key SAN genes, it was devoid of *NKX2-5* (*Figure 3D and E*), therefore closely resembling the transcriptional signature of the mouse *Tbx18$^+$/Nkx2-5$^-$* SAN-head region (*Wiese et al., 2009*). Cluster 5, on the other hand, revealed a transcriptional pattern found in the SAN tail, that is, *Tbx18$^-$/Tbx3$^+$/Nkx2-5$^+$* (*Wiese et al., 2009*; *Figure 3D and E*). The third cardiomyocyte cluster, cluster 4, expressed *TBX3* and lower levels of *ISL1* and *BMP4* (*Figure 3D*), but exhibited higher expression of atrial-associated genes, such as *NKX2-5*, *NPPA* (*Figure 3E*), *HAMP*, and *ADM* (*Litviňuková et al., 2020*; *Figure 3—figure supplement 1F*), demonstrating that these cells share characteristics of both pacemaker and atrial cells, identified as SAN-TZ cells in vivo (*Li et al., 2019*; *Goodyer et al., 2019*). Cluster 4 also revealed higher expression of *CPNE5* (*Figure 3—figure supplement 1F*), which is expressed throughout the entire cardiac conduction system and was found enriched in the transitional SAN region (*Goodyer et al., 2019*). Thus, we determined cluster 4 as SAN-TZ-like, cluster 5 as SAN-tail-like, and cluster 6 as SAN-head-like cells. Key genes differentially expressed in the SANCM subpopulations are presented in *Figure 3—figure supplement 2* and the complete list is provided as *Supplementary file 1*. The remaining two clusters from the SANCM group, clusters 7 and 8, were identified as sinus venosus-like cells and proepicardial cells, respectively. The *TBX18$^+$*, *SHOX2$^+$*, *BMP4$^+$*, *ISL1$^+$*, *TBX3$^-$*, *NKX2-5$^-$* expression pattern of cluster 7

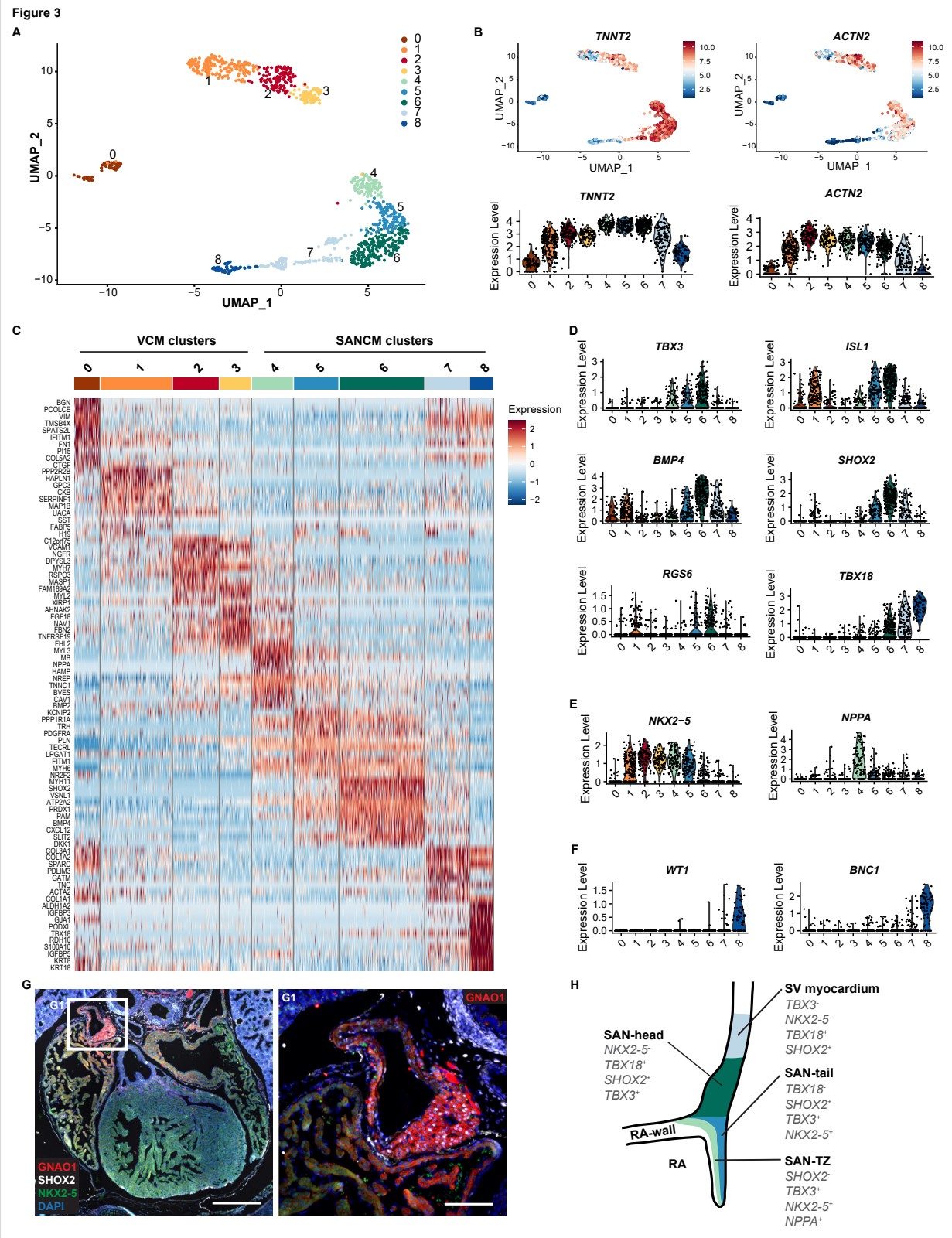

**Figure 3.** Single cell RNA-sequencing analysis of sinoatrial node-like cardiomyocyte (SANCM) and ventricular-like cardiomyocyte (VCM) cultures. (**A**) UMAP representation of single cell transcriptomes of SANCM and VCM at day 19 of differentiation. (**B**) UMAP feature plots and violin plots showing *TNNT2* and *ACTN2* expression in cell clusters. (**C**) Heatmap showing the top 10 differentially expressed genes in clusters at day 19 of differentiation. (**D–F**) Violin plots depicting expression of compact SAN-associated genes (**D**), SAN-TZ-associated genes (**E**), and proepicardial-associated genes

*Figure 3 continued on next page*

*Figure 3 continued*

(**F**). (**G**) Immunofluorescence staining of GNAO1 co-stained with SHOX2, NKX2-5, and DAPI in E17.5 embryonic mouse heart. Scale bar 500 μm. G1 is a zoom in of the marked SAN area. Scale bar 100 μm. (**H**) Schematic representation of the in vivo organization of the SV and SAN region during development. RA, right atrium; rvv, right venous valve; SAN, sinoatrial node; SV, sinus venosus; UMAP, uniform manifold approximation and projection. Also see *Figure 3—figure supplement 1* and *Figure 3—figure supplement 2*.

The online version of this article includes the following figure supplement(s) for figure 3:

**Figure supplement 1.** Single cell RNA-sequencing analysis of sinoatrial node-like cardiomyocyte (SANCM) and ventricular-like cardiomyocyte (VCM) cultures.

**Figure supplement 2.** Genes enriched in the different sinoatrial node-like cardiomyocyte (SANCM) subpopulations.

(*Figure 3D and E*) resembles the gene expression pattern of sinus venosus myocardium (*Christoffels et al., 2006*; *Blaschke et al., 2007*; *Espinoza-Lewis et al., 2009*; *Cai et al., 2003*; *Vicente-Steijn et al., 2010*). The expression pattern of cluster 8 was characterized by typical proepicardial markers such as *TBX18*, *KRT8*, *KRT18*, *WT1*, *BNC1* (*Lupu et al., 2020*; *Figure 3C and F*).

Besides well-established SAN genes, we also identified other markers such as *VSNL1* and *GNAO1*, which were specifically expressed in the SANCM clusters compared with the VCM clusters (*Figure 3—figure supplement 1H*). Visinin like 1 protein (VSNL1, also referred to as VILIP-1 or NVP-1) is a well-conserved $Ca^{2+}$-binding protein involved in various cellular signaling cascades (*Braunewell et al., 2009*) and has previously been identified in mouse, primate, as well as human SAN (*van Eif et al., 2019*; *Liang et al., 2021*). Immunofluorescence staining of E17.5 mouse heart confirmed robust expression of VSNL1 in the mouse SAN (*Figure 3—figure supplement 1I*). GNAO1 encodes the guanine nucleotide-binding protein G(o) subunit α, which is a part of the G-protein signal transducing complex (*Lambright et al., 1994*). We corroborated enriched expression of GNAO1 in the SAN of E17.5 mouse (*Figure 3G*). Both proteins were also expressed in the atria albeit to a lesser extent (*Figure 3G* and *Figure 3—figure supplement 1I*). In summary, hiPSC differentiation toward SANCM closely recapitulates the in vivo situation generating subpopulations with gene expression patterns resembling those of SAN-head, SAN-tail, and SAN-TZ cardiomyocytes (schematic in *Figure 3H*). Furthermore, small populations of co-differentiating sinus venosus-like and proepicardial-like cells alongside SANCM is reflective of shared developmental origins.

## hiPSC differentiation to SANCM recapitulates in vivo development

In order to gain a better understanding of the differentiation and specification process of hiPSCs to SANCM, we performed scRNA-seq at several stages during differentiation. At five additional time points (days 0, 4, 5, 6, and 10) (*Figure 4A*), cells were sorted and sequenced. A total of 3300 cells including the D19 SANCM population presented in *Figure 3* passed pre-processing and quality control. Unsupervised clustering was performed with the top 20 PCs, which identified 14 clusters. Two of the 14 clusters showed enriched expression of spike-in DNA/ERCCs (*Figure 4—figure supplement 1A*), indicating the amplification of ambient RNA and were therefore excluded from further analysis. The remaining clusters (comprised of 3103 cells) closely correlated with the time of collection, revealing that substantial transcriptional changes occur during the differentiation process in vitro (*Figure 4B* and *Figure 4—figure supplement 1B*). The expression of *TNNT2* and *ACTN2* steadily increased from day 5 (*Figure 4C*).

Next, we compared the gene expression profile (*Supplementary file 2*) of our time course dataset with a range of established stage-specific genes reflecting fate choices toward cardiomyocytes. From a pluripotent state at day 0 ($SOX2^+/NANOG^+/POU5F1^+$), the cells were directed toward germ layer specification with the majority of the cells (cluster D4_1) exhibiting a cardiac mesoderm-like profile expressing *EOMES*, *MESP1*, and *MESP2* (*Kitajima et al., 2000*; *Costello et al., 2011*; *Figure 4D and E*). A smaller endoderm-like population was also identified on day 4 (cluster D4_2), based on the specific expression of *FOXA2* and *SOX17* (*Tosic et al., 2019*; *Figure 4—figure supplement 1C*). After 24 hr with SAN specification medium (day 5, D5), we identified a gene expression pattern, characteristic for posterior cardiac progenitors ($HOXA1^+/NR2F2^+/TBX5^+$) (*Figure 4F*; *Bertrand et al., 2011*; *Stefanovic et al., 2020*). The first onset of *TBX18* expression was observed at day 6 (D6) of differentiation (*Figure 4G*), a transcription factor marking sinus venosus progenitors (*Mommersteeg et al., 2010*). Similarly, *TBX3* was expressed in a cell fraction collected on day 6 (D6) (*Figure 4G*).

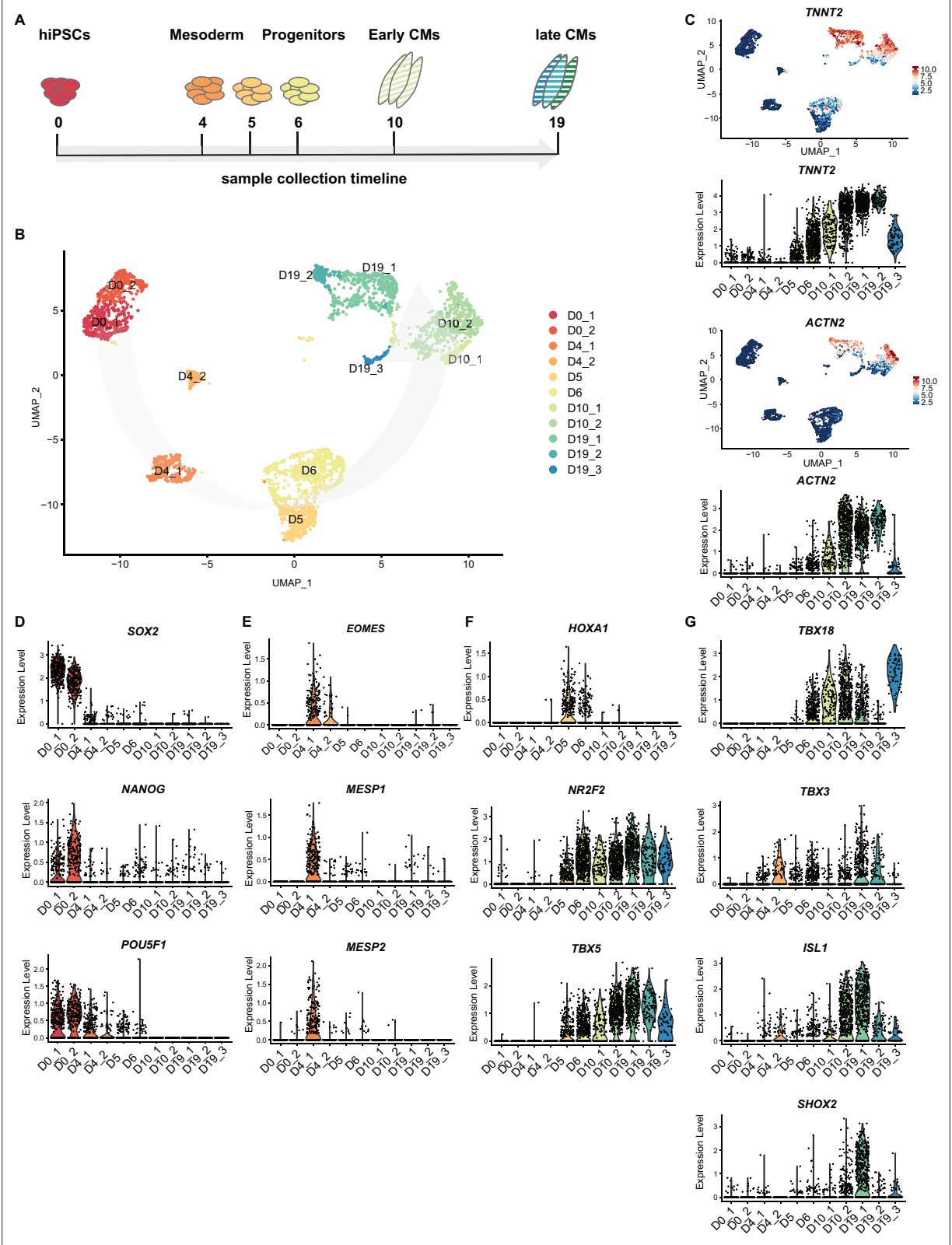

**Figure 4.** Time course single cell RNA-sequencing of SANCM. (**A**) Timeline of hiPSC differentiation to SANCM representing sample collection time points. (**B**) UMAP representation of single cell transcriptomes collected at different time points throughout differentiation from hiPSC to SANCM. Arrow indicates course of differentiation. (**C**) UMAP feature plots and violin plots showing *TNNT2* and *ACTN2* gene expression at different stages of SANCM differentiation. (**D–H**) Violin plots of pluripotency genes (**D**), mesodermal genes (**E**), posterior cardiac progenitor genes (**F**), proepicardial genes (**G**), and

*Figure 4 continued on next page*

*Figure 4 continued*

SAN-associated transcription factor genes (**H**). hiPSCs, human induced pluripotent stem cells; CPC, cardiac progenitor cells; CMs, cardiomyocytes; UMAP, uniform manifold approximation and projection; SANCM, sinoatrial node-like cardiomyocyte. Also see *Figure 4—figure supplement 1* and *Figure 4—figure supplement 2*.

The online version of this article includes the following figure supplement(s) for figure 4:

**Figure supplement 1.** Time course single cell RNA-sequencing of sinoatrial node-like cardiomyocyte (SANCM).

**Figure supplement 2.** Time course RT-qPCR of sinoatrial node (SAN)-subpopulation markers.

Early-stage differentiated cells at day 10 (D10) formed two clusters. Cluster D10_1 representing a less mature state compared with cluster D10_2 according to *TNNT2* and *ACTN2* expression (*Figure 4C*). Furthermore, cluster D10_1 is partially composed of cells collected on day 19 (D19) of differentiation identified as sinus venosus-like cells (cluster 7) in *Figure 3* and *Figure 4—figure supplement 1D*, suggesting that a fraction of cells was halted during differentiation. Notably, the expression of well-known SAN-associated transcription factors *ISL1* and *SHOX2* was observed from D10 onward (cluster D10_2) (*Figure 4G*).

On D19 of differentiation, three separate clusters were identified comprising SANCM subpopulations (D19_1 and D19_2) and proepicardial-like cells (D19_3), as described in detail in *Figure 3*. Interestingly, *PDPN*, reported to be expressed both in the SAN and the epicardium in the mouse heart (*Gittenberger-de Groot et al., 2007*), was found exclusively in the $ALDH1A2^+/WT1^+$ proepicardial-like population (D19_3) and not in SANCM on day 19 (*Figure 4—figure supplement 1E*). List of genes differentially expressed in each cluster are provided in *Supplementary file 2*.

To complement our single cell analysis with additional time points collected during the differentiation process, we performed RT-qPCR for key SAN markers including genes differentially expressed in the different subpopulations. As also identified by scRNA-seq, *TBX5* and *TBX18* appear early in the differentiation process (*Figure 4F, G* and *Figure 4—figure supplement 2*). *SHOX2* expression begins on day 8 and gradually increases over time. With the exception of *FLRT3*, which is already expressed at day 6, most subpopulation markers begin to be expressed from day 8 onward (*Figure 4—figure supplement 2B, C*). Taken together, our findings reveal that the in vitro differentiation described here is a valuable model to study the earliest steps of pacemaker cell specification, overcoming the limitation of accessibility to comparable in vivo developmental stages.

## WNT signaling mediates the divergence of myocardial and proepicardial lineages

scRNA-seq of SANCM revealed the presence of different SAN subtypes, such as SAN-head, SAN-tail, and SAN-TZ, which co-differentiate with a small population of proepicardial-like cells (Epi). To gain insight into the developmental ontogeny of these cell types, we used URD (*Farrell et al., 2018*). URD reconstructs transcriptional trajectories based on user-defined origin (root) and end points (tips). We assigned the cardiac mesoderm stage (day 4) as the root and the distinct subclusters identified on day 19 (*Figure 3*), that is, SAN-head, SAN-tail, SAN-TZ, and proepicardial cells as the tips, resulting in a pseudotime tree consisting of six main segments (*Figure 5A*). Sinus venosus-like cells were excluded as a tip since it partially clustered with progenitors of day 10 and is a cell type independent of the SAN niche (*Figure 4—figure supplement 1D*). Cells from day 5, day 6, and a fraction of day 10 were located near the root of the tree in segment 1, constituting a common progenitor pool. From segment 1, the pseudotime tree branches off into two lineages, the proepicardial branch (segment 2) and the myocardial branch (segment 3). The proepicardial branch contained cells collected on day 10 as well as day 19, whereas the myocardial branch primarily consisted of cells collected on day 10. While most myocardial cells at day 10 were present in segment 3, a small fraction appeared committed to SAN-TZ lineage (segment 4). SAN-tail (segment 5) and SAN-head (segment 6) were assigned later pseudotimes and only contained cells from day 19. Similar findings were obtained using a second trajectory inference method, Slingshot (*Street et al., 2018*; *Figure 5—figure supplement 1A-C*).

The ordering of cell populations in the trajectory tree suggests that cells on day 5 and day 6 could potentially give rise to both the myocardial and proepicardial lineages. The first divergence was only apparent at day 10 (*Figure 5A*) with the majority of the cells directed toward the myocardial lineage whereas a small population branched off toward the proepicardial lineage. Accordingly,

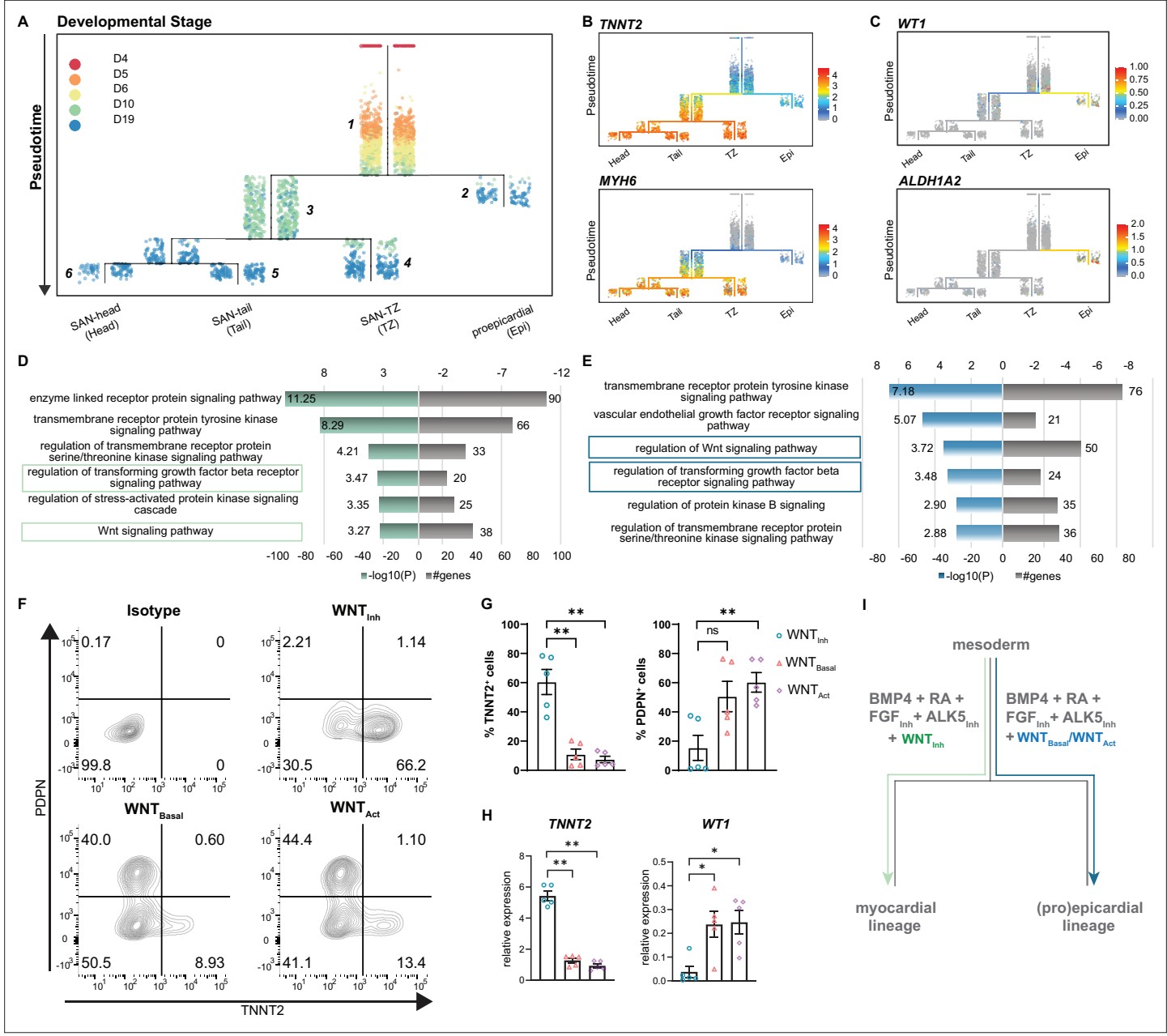

**Figure 5.** Reconstruction of single cell trajectories. (**A**) URD trajectory tree starts at late mesoderm stage (day 4) and proceeds to terminally differentiated cell clusters identified on day 19. Colors correspond to the time point of cell collection. (**B**) Expression of *TNNT2* and *MYH6* marking the myocardial lineage (**C**) Expression of *WT1* and *ALDH1A2* marking the proepicardial lineage in the trajectory tree. (**D–E**) Representative gene ontology (GO) terms based on differentially expressed genes between the common progenitor, segment 1, and the myocardial branch, segment 3, (**D**) or the proepicardial branch, segment 2 (**E**) . (**F**) Representative contour plots and (**G**) summarized data demonstrating percentage of TNNT2[+] and PDPN[+] cells in baseline condition containing WNT inhibitor, XAV (WNT$_{Inh}$), excluding WNT inhibitor, XAV (WNT$_{Basal}$), and addition of WNT activator, CHIR (WNT$_{Act}$). N=5 independent differentiations. Error bars represent s.e.m., Kruskal-Wallis, post hoc Mann-Whitney U test: p<0.005 (**). (**H**) RT-qPCR demonstrating the expression of cardiomyocyte gene *TNNT2* and the proepicardial gene *WT1* in WNT$_{Inh}$, WNT$_{Basal}$, and WNT$_{Act}$ conditions. N=5 independent differentiations; corrected to GEOMEAN of reference genes *RPLP0* and *GUSB*. Error bars, s.e.m. Kruskal-Wallis, post hoc Mann-Whitney U test: p<0.05 (*), p<0.005 (**). (**I**) Schematic representation of divergence of myocardial and proepicardial lineages from a common progenitor. Also see *Figure 5— figure supplement 1*.

The online version of this article includes the following figure supplement(s) for figure 5:

**Figure supplement 1.** Reconstruction of single cell trajectories and the role of TGFβ modulation on pacemaker versus proepicardial differentiation.

cardiomyocyte genes such as *TNNT2* and *MYH6* were selectively expressed in the myocardial branch (*Figure 5B*), and proepicardial genes such as *WT1* and *ALDH1A2* were enriched in the proepicardial branch (*Figure 5C*). Thus, day 10 of differentiation appears to be a critical branching point for myocardial and proepicardial cell fates driven by BMP and RA.

In order to identify the key players that regulate myocardial versus proepicardial cell fate, we performed gene ontology (GO) analysis on differentially expressed genes between the common progenitor (segment 1) and the myocardial branch (segment 3) or the common progenitor (segment 1) and the proepicardial branch (segment 2) (*Figure 5A*). GO term analysis identified signaling pathways potentially involved in myocardial versus proepicardial divergence (*Supplementary file 3*). As both groups contained a number of genes implicated in TGFβ and WNT signaling (*Figure 5D and E*), we tested the impact of manipulating these signaling pathways on myocardial versus proepicardial fate specification. The standard SANCM differentiation cocktail contains the ALK5 inhibitor (SB431542) included to offset any effects of BMP4 on TGFβ signaling (*Birket et al., 2015*; *Protze et al., 2017*). To allow active TGFβ signaling, we excluded the ALK5 inhibitor (w/o ALK5$_{Inh}$) from the SANCM differentiation cocktail. Our results show that active TGFβ signaling does not alter myocardial versus proepicardial cell fate, as determined by the expression of TNNT2 marking cardiomyocytes, and PDPN marking proepicardial cells, in the resulting population (*Figure 5—figure supplement 1D and E*).

Next, we tested the role of WNT signaling in myocardial versus proepicardial branching. From the SANCM differentiation cocktail containing the WNT inhibitor, XAV939 (WNT$_{Inh}$) (*Figure 1A*), we either removed the WNT inhibitor (WNT$_{Basal}$) or replaced it with the WNT agonist, CHIR (WNT$_{Act}$). Applying the standard cocktail containing the WNT inhibitor resulted in an average of 60% TNNT2$^+$ cells with a small side population of 10% PDPN$^+$ cells (*Figure 5F and G*). Strikingly, removal of the WNT inhibitor strongly compromised the percentage of cardiomyocytes (~10% TNNT2$^+$), whereas percentage of PDPN$^+$ cells increased. Addition of a WNT agonist had a similar effect although it did not further enhance the percentage of proepicardial cells. RT-qPCR confirmed a proepicardial-like gene expression in WNT$_{Basal}$ and WNT$_{Act}$ conditions evidenced by higher expression of *WT1* and lower expression of *TNNT2* mRNA compared with WNT$_{Inh}$ (*Figure 5H*). These findings demonstrate that in the presence of active WNT signaling, BMP and RA steer common progenitors toward the proepicardial fate and that inhibition of WNT signaling is crucial for their differentiation toward the myocardial lineage (*Figure 5I*).

## Diversification between the myocardial SAN subpopulations involves WNT and TGFβ signaling

To better understand the mechanisms implicated in the specification of SAN subpopulations, we looked at transcriptional changes between common progenitor state (segment 3) and day 19 SANCM subpopulations (segments 4–6). Ordering of cells in the trajectory tree in *Figure 5A* suggests that a large majority of myocardial cells remain uncommitted at day 10 and specification toward SAN-head and SAN-tail cells only occurs after day 10. GO term analysis of differentially expressed genes between the common myocardial progenitor at day 10 (segment 3) and each SANCM subtype of day 19 (segments 4–6) revealed enrichment of several WNT signaling modulators, such as *DKK1*, *WNT5A*, *SFRP1*, and *APP*, primarily in the SAN-head branch (segment 6) (*Figure 6A* and *Supplementary file 4*). Whilst DKK1 is an inhibitor of canonical WNT signaling, WNT5A is a non-canonical WNT ligand. Therefore, we posited that inhibition of canonical WNT signaling may enhance differentiation to SAN-head-like cells. To test this assumption, we treated SANCM cultures with XAV 939 from day 10 to day 17, following the findings from the trajectory tree. We determined the effect of this treatment on SANCM by assessing the expression of genes specific to or enriched in individual SANCM fractions, such as *SHOX2*, *VSNL1*, *NTM*, and *FLRT3* for SAN-head, *KCNIP2* for SAN-tail, and *NKX2-5*, *NPPA*, and *CPNE5* for SAN-TZ (*Figure 3—figure supplement 1F* and *Figure 3—figure supplement 2A and B*). Our results show that inhibition of canonical WNT signaling from days 10 to 17 significantly increased expression of SAN-head-enriched genes, such as *SHOX2*, *NTM*, and *VSNL1*, and a trend for higher expression in *FLRT3* (*Figure 6B*) but did not influence the expression of SAN-tail or SAN-TZ genes (data not shown).

Similarly, members of the TGFβ/BMP signaling pathway were preferentially expressed in SANCM subpopulations at day 19. These included ligands *BMP4* (SAN-head) and *BMP2* (SAN-TZ), as well as genes involved in TGFβ/BMP signaling such as *HTRA1* (SAN-head; SAN-tail) and *FBN2* (SAN-TZ) (*Figure 6C* and *Supplementary file 4*), implicating this pathway in differentiation toward SAN subpopulations. Moreover, the percentage of TNNT2$^+$ cells in the w/o ALK5$_{Inh}$ condition were unaffected (*Figure 5—figure*

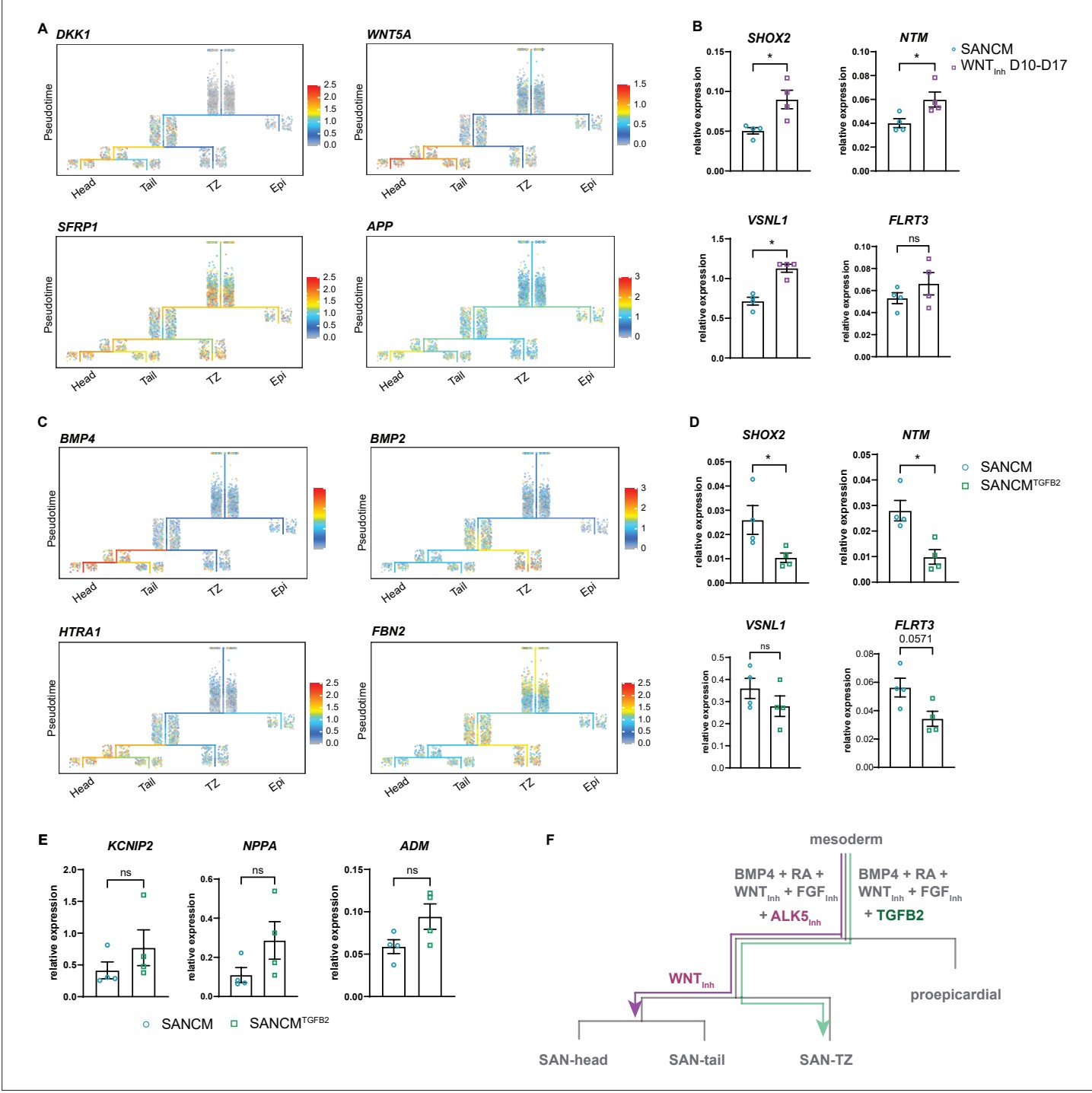

**Figure 6.** Diversification of sinoatrial node (SAN)-head, SAN-tail, and SAN-TZ subpopulations. (**A**) Expression of WNT signaling pathway members in the trajectory tree. (**B**) RT-qPCR of genes enriched in the SAN-head lineage upon prolonged WNT signaling inhibition (WNT$_{Inh}$ D 10–17). N=4 independent differentiations; corrected to GEOMEAN of reference genes *RPLP0* and *GUSB*. Error bars, s.e.m. Mann-Whitney U test: p<0.05 (*). (**C**) Expression of TGFβ signaling pathway members in the trajectory tree. (**D–E**) RT-qPCR of genes enriched in the SAN-head lineage (**D**) and SAN-tail/-TZ lineage (**E**) upon supplementation with TGFB2 (SANCM$^{TGFB2}$) during differentiation. N=4 independent differentiations; corrected to GEOMEAN of reference genes *RPLP0* and *GUSB*. Error bars, s.e.m. Mann-Whitney U test: p<0.05 (*). (**F**) Schematic representation of the diversification of the various SANCM subpopulations from a common myocardial progenitor.

supplement 1D and E), suggesting that basal TGFβ signaling does not affect myocardial specification itself. These observations led us to evaluate the effect of TGFβ signaling on the identity of pacemaker subpopulations, which was achieved by replacing the TGFβ signaling inhibitor SB531542 with the TGFβ ligand TGFB2 from day 4 to day 6 of SANCM differentiation (SANCM^TGFB2). RT-qPCR revealed a downregulation of SAN-head-associated genes, such as *SHOX2* and *NTM*, and a trend for reduced expression of *VSNL1* and *FLRT3*, even though not statistically significant (*Figure 6D*). Furthermore, we observed a higher expression of SAN-tail-associated gene *KCNIP2* and SAN-TZ-associated genes, *NPPA* and *ADM*, in TGFB2 supplemented differentiations, even though not statistically significant (*Figure 6E*). Taken together, these findings underscore a stage-specific role for WNT and TGFβ signaling in differentiation toward specific SANCM subpopulations (*Figure 6F*).

## TGFβ signaling promotes differentiation toward transitional cells

Because SANCMs showed a shift in marker expression toward SAN-TZ cells in SANCM^TGFB2 condition, we next asked whether and to what extent the composition of SANCM subpopulations would change in this condition. We performed scRNA-seq of day 19 SANCM^TGFB2 cultures and included day 19 SANCM as well as hiPSC-derived atrial cells (ACM) (*Devalla et al., 2015*; *Li et al., 2019*) as gene expression characteristics of SAN-TZ cells are expected to overlap with both these cell types. Unsupervised clustering of day 19 cells collected from SANCM, SANCM^TGFB2, and ACM identified eight different clusters (*Figure 7—figure supplement 1A and B*). Based on *TNNT2* and *ACTN2* expression, non-cardiomyocyte clusters were excluded (*Figure 7—figure supplement 1C*), which resulted in a total of five cardiomyocyte clusters (*Figure 7A*). Based on the expression profiles (*Figure 7A*), we determined that cluster 0 contains atrial cardiomyocytes and cluster 1 is composed of pacemaker cells. Furthermore, cluster 3 consisted of sinus venosus-like cells. The identity of cluster 4 could not be discerned but the expression of *IRX5* suggests these are a subpopulation of atrial cells as reported in vivo (*Bosse et al., 2000*; *Gaborit et al., 2012*). Lastly, cluster 2 expressed several atrial genes such as *NKX2-5*, *NPPA*, *HAMP* (*Figure 7—figure supplement 1D*), but also shared similarities with pacemaker cells in cluster 1 (*Figure 7—figure supplement 1E*). In addition, this cluster expressed *CPNE5* suggesting that they are transitional cells (*Goodyer et al., 2019*; *Figure 7—figure supplement 1D*).

Next, we assessed the origin of cells present in the five clusters (*Figure 7A*). Cells from SANCM differentiations were present in cluster 1 (pacemaker cells) and cluster 2 (transitional cells), and cells from ACM differentiations were present in cluster 0 (atrial cells) and cluster 2 (transitional cells). Consistent with the expression analysis (*Figure 6D and E* and *Supplementary file 5*), cells from SANCM^TGFB2 were present mainly in cluster 2 containing transitional cells. In order to identify which SANCM subpopulations are present in cluster 2, we highlighted the cells previously annotated as SAN-head, SAN-tail, and SAN-TZ in *Figure 3* in the cluster analysis comparing SANCM, SANCM^TGFB2, and ACM (*Figure 7D*). This visualization confirmed that SAN-TZ cells and SANCM^TGFB2 cells clustered together in cluster 2, whereas SAN-head and SAN-tail cells independently formed cluster 1 (*Figure 7D*). In essence, TGFB2 supplementation during SANCM differentiation steers cells toward a transitional phenotype distinct from SAN-head, SAN-tail, and atrial cardiomyocytes.

Finally, we characterized the electrophysiological properties of transitional cells obtained from SANCM^TGFB2 cultures by single cell patch clamp. Representative AP traces are shown in *Figure 7E*. Single cell patch clamp revealed longer cycle lengths (792.3±49.1 ms, mean ± s.e.m., n=8) compared with SANCM (552.8±29.1 ms, mean ± s.e.m., n=6) (*Figure 7F*). Consistent with a more atrial-like phenotype, the MDP was more negative in SAN-TZ cells (–69.1±1.8 mV) compared with SANCM (–60.1±3.3 mV), but not as negative as in ACM (–77.4±1.4 mV) (data from Li et al., in revision) (*Figure 7F* and *Figure 7—figure supplement 1F*). Furthermore, upstroke velocities (Vmax) were higher in SAN-TZ cells (*Figure 7G*) and were comparable to ACM (*Figure 7—figure supplement 1F*). APA as well as APD (APD20, APD50, and APD90) did not differ between the two groups (*Figure 7G*). In sum, we identified a critical role for TGFβ signaling in the specification of SAN-TZ subpopulation with gene expression and electrical characteristics intermediate to that of SAN and atrial cardiomyocytes.

## Discussion

The developmental ontogeny of the SAN is poorly understood. Here, we aimed to study the differentiation and diversification of the human SAN in vitro. Directed differentiation of hiPSCs to cardiomyocytes was achieved using a two-step approach, wherein hiPSCs were first directed toward mesoderm,

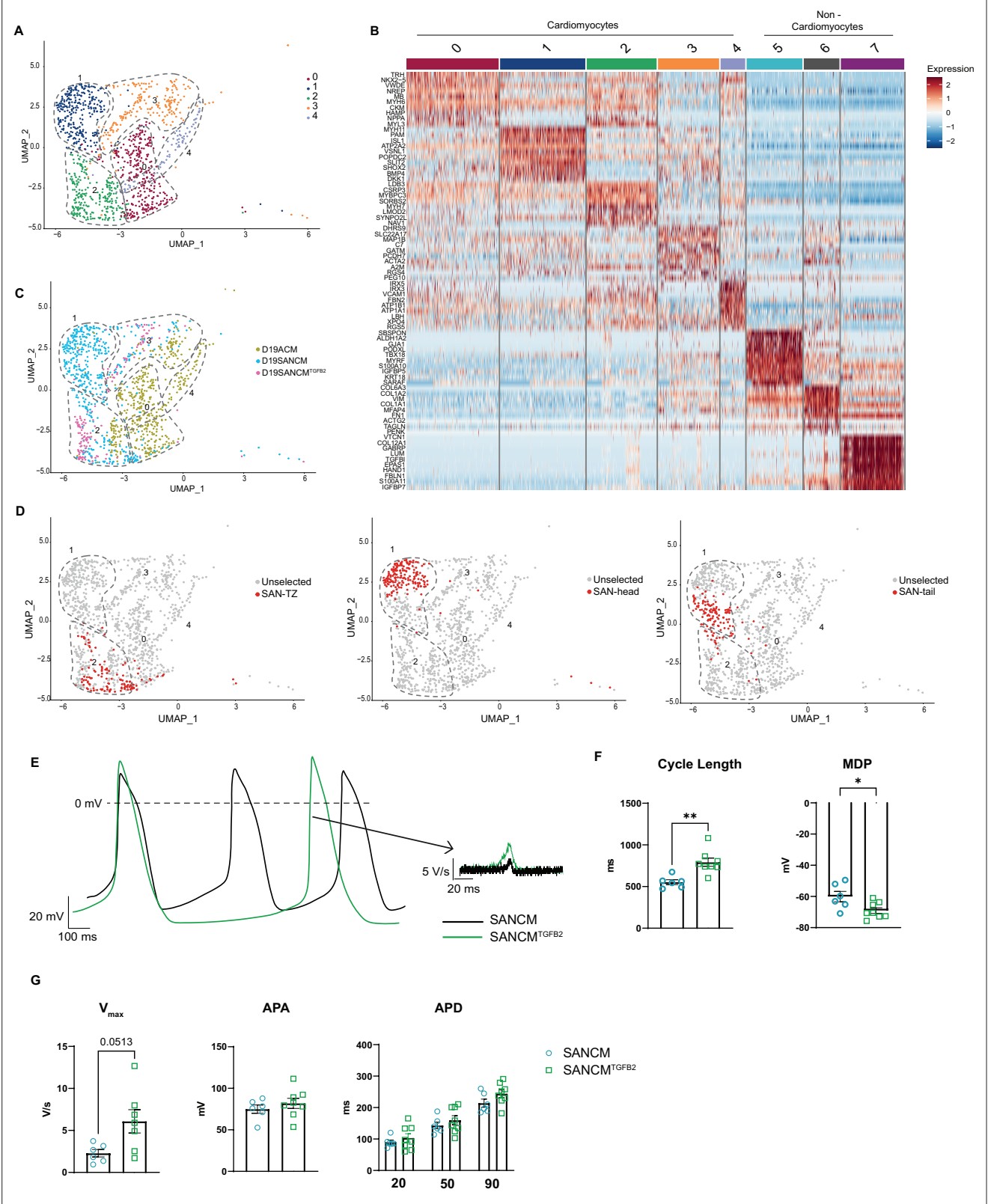

**Figure 7.** TGFβ signaling promotes differentiation toward sinoatrial node (SAN)-TZ cells. (**A**) Uniform manifold approximation and projection (UMAP) representation of single cell transcriptomes of cardiomyocyte clusters derived from SANCM, SANCM$^{TGFB2}$, and ACM differentiation at day 19. (**B**) Heatmap depicting the top 10 differentially expressed genes in each cluster. (**C**) UMAP showing the distribution of cells from different conditions in clusters 0–4. (**D**) UMAP highlighting previously annotated SAN subpopulation clusters (SAN-TZ, SAN-head, and SAN-tail) (**Figure 3**) in the cluster

*Figure 7 continued on next page*

Figure 7 continued

analysis comparing SANCM, SANCM[TGFB2], and ACM. (**E**) Representative traces of spontaneous APs of day 19 SANCM (black) and SANCM[TGFB2] (green). (**F–G**) Cycle length and MDP (**F**), Vmax, APA, and APD20, and APD50, and APD90 (**G**) of SANCM and SANCM[TGFB2] at day 19 of differentiation. N=6–8 cells. Error bars, s.e.m. Mann-Whitney U test: p<0.05 (*), p<0.005 (**), p<0.0001 (****). MDP, maximal diastolic potential; APA, action potential amplitude; Vmax, upstroke velocity; APD20, APD50, APD90, AP duration at 20%, 50%, 90% repolarization, respectively. Also see *Figure 7—figure supplement 1* and related source data file *Figure 7—source data 1*.

The online version of this article includes the following source data and figure supplement(s) for figure 7:

**Source data 1.** Active TGFβ signaling promotes SAN-TZ cell differentiation.

**Figure supplement 1.** TGFβ signaling promotes differentiation toward sinoatrial node (SAN)-TZ cells.

which was further steered toward a cardiac fate by inhibition of WNT signaling. This standard approach generated cardiomyocytes with a ventricular-like signature. Alongside inhibition of WNT signaling, addition of BMP4 and RA as well ALK5 inhibitor, and FGF inhibitor, at the cardiac mesoderm stage resulted in cardiomyocytes with SAN-like profile as reported before (*Protze et al., 2017*). Whereas the identity of SAN cells obtained in this previous study appears to be predominantly SAN-head-like inferred by NKX2.5[-] and TNNT2[+] expression, we found the presence of multiple cell types that develop at the inflow tract of the heart in our in vitro cultures. Whether this is due to differences in the methods used for mesoderm induction or due to different culture conditions (3D versus 2D) is not known. Nevertheless, obtaining various human SAN subpopulations is valuable for both in vitro and in vivo applications.

Current differentiation protocols for SANCM result in 25–50% NKX2-5[-] SANCM resembling SAN-head (or even a fraction of sinus venosus-like cells) while no data is available with regard to the presence of NKX2.5[+] SAN-tail and SAN-TZ cells (*Protze et al., 2017*; *Ren et al., 2019*). We found that 54% of the entire SANCM pool exhibited the expression pattern described in previous studies (*NKX2-5[-]/TNNT2[+]*). A fraction of these *NKX2-5[-]* cardiomyocytes (18% of total SANCM pool) resembled the gene expression pattern of sinus venosus myocardium, characterized by lower expression of sarcomeric genes such as *TNNT2* and *ACTN2*, presence of *SHOX2* and absence of *TBX3*. Importantly, 36% of the total SANCM pool revealed a gene expression pattern similar to the SAN-head (*NKX2-5[-]/TBX18[+]/TBX3[+]*), whereas 19% resembled a SAN-tail like phenotype (*NKX2-5[+]/TBX18[-]/TBX3[+]*) and 17%, a transitional cell-like phenotype (*NKX2-5[+]/CPNE5[+]/TBX3[+]*) (*Wiese et al., 2009*; *Sizarov et al., 2011*; *Goodyer et al., 2019*).

Time course analysis in this study indicated that cells at day 5, that is, 24 hr after the addition of pacemaker differentiation factors, correspond to posterior second heart field progenitors as identified by the expression of *HOX* and *TBX* genes. Furthermore, cells at differentiation day 8 appear to correspond to E8.5–E9.5 of mouse development based on the expression of *SHOX2* and *ISL1* (*Liang et al., 2015*; *Espinoza-Lewis et al., 2009*). These results indicate that in vitro differentiation recapitulates development of pacemaker cells in vivo. Our findings further demonstrate that the expression of SAN subpopulation markers begins as early as day 6 (*FLRT3*) with most others expressed from day 8 onward. Our data thus provides insights into the earliest steps of pacemaker specification and is a valuable model to study early events typically with limited access in animal models.

To gain insight into the origin and diversification of the cell types in the SANCM group, we applied trajectory inference analysis. Our data revealed early divergence between the myocardial SAN and proepicardial populations in line with previous reports that identified a *Tbx18[+]* common progenitor for these lineages (*van Wijk et al., 2009*; *Kruithof et al., 2006*). Besides SAN development, BMP and RA signaling are also implicated in the development of the epicardium and a crosstalk with WNT signaling has been postulated (*Wiesinger et al., 2021*). Our results show that WNT signaling in fact determines the bifurcation of myocardial and proepicardial cell fates. Excluding the WNT inhibitor in the presence of BMP4 and RA diminished the myocardial population and enriched the proepicardial population. Our findings also correlate with a previous study, which described the generation of proepicardial cells under similar experimental conditions (*Guadix et al., 2017*). Comparably, the PDPN[+] proepicardial population in the study of Guadix et al. was prominent in culture condition with BMP4 and RA, which strongly decreased with the addition of a WNT inhibitor. Active WNT signaling therefore seems pertinent for epicardial cell differentiation, as indicated in several in vitro differentiation studies (*Witty et al., 2014*; *Iyer et al., 2015*; *Bao et al., 2016*; *Zhao et al., 2017*). Consistent with our results, intrinsic WNT activity is sufficient for epicardial differentiation by BMP4 and RA (*Iyer et al., 2015*;

*Guadix et al., 2017*). Similarly, activation of WNT signaling at the *NKX2.5*+ cardiac progenitor stage resulted in SANCMs as well as a non-cardiomyocyte population, which exhibited an epicardial-like phenotype (*Ren et al., 2019*).

Our dataset provided further insight into the developmental trajectory of SANCM. A proportion of the cardiac progenitor cells collected on day 10 were already committed to the SAN-TZ lineage, whereas none of these progenitors appeared determined to SAN-head or SAN-tail lineages, indicating that SAN-TZ cell specification occurs earlier. The order of differentiation of the components of the mouse sinus venosus and SAN have been analyzed in detail. During mouse caudal heart development, *Tbx18*- posterior second heart field progenitors first form the inflow tract of the myocardial heart tube, which differentiate into atrial cardiomyocytes. Subsequently, *Tbx18*+ progenitors differentiate to cardiomyocytes and form the SAN and sinus venosus components in the order of their future anatomical position from proximal to distal of the atrial myocardium. Thus, the *Tbx18*+ progenitors first form the *Tbx3*+ transitional pacemaker cells, directly followed by the *Tbx3*+ SAN tail, the *Tbx3*+ SAN-head, and finally the *Tbx3*- sinus venosus myocardium of the superior caval vein (*Christoffels et al., 2006*; *Mommersteeg et al., 2007*; *Wiese et al., 2009*; *Mommersteeg et al., 2010*; *Mohan et al., 2018*). It is therefore interesting to note that the developmental trajectory of the SANCM cells in vitro recapitulate this temporal aspect of in vivo mouse SAN development. A fraction of the cells collected on day 19 formed a small common segment before separating into the SAN-head or the SAN-tail tips, indicating that differentiation of the cell types may not yet be complete. Nevertheless, our data shows that SAN-head, SAN-tail, and SAN-TZ originate from a common progenitor, which under the influence of various signaling pathways diversify into these subpopulations.

Consistent with our findings, TGFβ/BMP signaling mediators have been found enriched in the embryonic SAN (*Vedantham et al., 2015*; *van Eif et al., 2019*; *Li et al., 2019*; *Goodyer et al., 2019*), which is maintained in adulthood (*Linscheid et al., 2019*; *Brennan et al., 2020*). Even though the exact role of TGFβ/BMP signaling during SAN development is not known, it has been proposed to be involved in recruitment of proepicardial cells and remodeling of interstitium in the SAN niche (*Easterling et al., 2021*). Comparably, WNT signaling has been described as a critical cue for SAN development (*Bressan et al., 2013*; *Ren et al., 2019*). Our results revealed a role for WNT and TGFβ signaling in enhancing gene signature pertaining to SAN-head and SAN-TZ cells, respectively. We further focused on characterizing the role of TGFβ and its effect on SAN subpopulations as knowledge pertaining to SAN-TZ cells is limited both in vitro and in vivo. We identified that TGFβ signaling in combination with WNT and RA signaling drives differentiation toward SAN-TZ cells exclusively. Molecular and electrophysiological characterization of these cells demonstrated that they are a distinct population, which share features with both pacemaker and atrial cells. Our findings thus identify a method to specifically steer differentiation toward SAN-TZ cells.

Principles of stage-specific manipulation of signaling pathways described in this study can be applied to other pluripotent stem cell lines including patient-specific lines to obtain desired cell fractions to create physiologically relevant in vitro models of the pacemaker niche. Such efforts will enable modeling of complex diseases such as SAN exit block, which occurs due to impaired impulse propagation to the atria and is thought to result from dysfunctional transitional cells (*Li et al., 2019*). Generation of SAN subpopulations and a better understanding of their importance in impulse generation and propagation is also crucial for developing novel treatment strategies including cell-based approaches (*Komosa et al., 2021*). Incorporating SAN subpopulations in the design of biomimetic cell constructs would permit the evaluation of optimal configurations that effectively regenerate the dysfunctional pacemaker tissue.

# Materials and methods

**Key resources table**

| Reagent type (species) or resource | Designation | Source or reference | Identifiers | Additional information |
|---|---|---|---|---|
| Cell line (*Homo sapiens*) | hiPSC line (female) | iPSC core facility of Leiden University Medical Center | LUMC0099iCTRL04 | https://hpscreg.eu/cell-line/LUMCi004-A |
| Antibody | Anti-cTNT (rabbit polyclonal) | Abcam | Ab45932 | (1:1000) |

*Continued on next page*

*Continued*

| Reagent type (species) or resource | Designation | Source or reference | Identifiers | Additional information |
|---|---|---|---|---|
| Antibody | Anti-ACTN2 (mouse monoclonal) | Sigma | A7811 | (1:800) |
| Antibody | Anti-SHOX2 (mouse monoclonal) | Abcam | ab55740 | (1:200) |
| Antibody | Anti-MYL2 (rabbit polyclonal) | Abcam | 79935 | (1:200) |
| Antibody | Anti-ISL1 (goat polyclonal) | Neuromics | GT15051 | (1:200) |
| Antibody | Anti-HCN4 (rabbit polyclonal) | Merck Millipore | AB5808-200uL | (1:250) |
| Antibody | Anti-NKX2-5 (goat polyclonal) | LabNed | LN2027081 | (1:150) |
| Antibody | Anti-GNAO1 (rabbit polyclonal) | Protein Tech Group | 12635-1-AP | (1:150) |
| Antibody | Anti-VSNL1 (rabbit polyclonal) | Abbexa | abx007357 | (1:450) |
| Antibody | Anti-cTNT -REAfinity (recombinant human; APC) | Miltenyi Biotec | 130-120-403 | (1:50) |
| Antibody | Anti-Podoplanin (rat monoclonal; Alexa Fluor 488) | Biolegend | 337005 | (1:20) |
| Peptide, recombinant protein | Activin-A | Miltenyi Biotec | #130-115-012 | Human, premium grade |
| Peptide, recombinant protein | BMP4 | R&D Systems | #314BP-010/CF | Recombinant human protein, carrier-free |
| Peptide, recombinant protein | TGFB2 | R&D Systems | #302-B2-002/CF | Recombinant human protein |
| Chemical compound, drug | CHIR99021 | Axon Medchem | #1386 | |
| Chemical compound, drug | XAV939 | Tocris Bioscience | #3748/10 | |
| Chemical compound, drug | SB431542 | Tocris Bioscience | #1614 | |
| Chemical compound, drug | PD173074 | Selleck Chemicals | #1264 | |
| Chemical compound, drug | Retinoic acid | Sigma | #R2625 | |
| Software, algorithm | Seurat V3/V4 | *Stuart et al., 2019* (V3) *Hao et al., 2021* (V4) | https://github.com/satijalab/seurat/ | |
| Software, algorithm | URD | *Farrell et al., 2018* | https://schierlab.biozentrum.unibas.ch/urd | |
| Software, algorithm | GraphPad Prism version 9.1.0 | GraphPad Software, San Diego, CA | https://www.graphpad.com/ | |
| Other | mTESR1 | Stem Cell Technologies | #5850 | iPSC Maintenance Media |
| Other | Matrigel | Corning | #356234 | Substrate for iPSC culture |
| Other | ×1 TryPLE Select | Thermo Fisher Scientific | #12563011 | Cell dissociation reagent |

## Maintenance of hiPSC lines and differentiation to cardiomyocytes

hiPSC line LUMC0099iCTRL04 used in this study was generated by the iPSC core facility of Leiden University Medical Center following due protocols for informed consent and use of these cells for research purposes. The cell line is registered in Human Pluripotent Stem Cell Registry (https://hpscreg.eu/cell-line/LUMCi004-A).

hiPSCs were maintained in mTESR1 medium (Stem Cell Technologies, #5850) on growth factor reduced Matrigel (Corning, #356234) at 37°C with 5% $CO_2$ and passaged once a week. Cells are tested for mycoplasma (Lonza, #LT07-218) contamination at least once a month.

For cardiac differentiation, cells were seeded at a density of 2.5–3×10$^4$ cells/cm$^2$. Differentiation was induced when cells reached 80–90% confluency using BPEL medium (*Ng et al., 2008*) supplemented with 20 ng/mL Activin-A (Miltenyi Biotec, #130-115-012), 20 ng/mL BMP4 (R&D Systems, #314 BP-010/CF), and 1.5 μmol/L CHIR99021 (Axon Medchem, #1386). Three days after initiation, medium was replaced with BPEL containing 5 μmol/L XAV939 (Tocris Bioscience, #3748/10). For SANCM differentiation, 5 μmol/L XAV939, 2.5 ng/mL BMP4, 5 μmol/L SB431542 (Tocris, #1614), 250 nmol/L RA (Sigma, #R2625-50MG), and 250 nmol/L PD173074 (Selleck Chemicals, #1264) were added on day 4. Differentiation medium was replaced with BPEL medium after 48 hr (SANCM) or 96 hr (VCM) and cells refreshed every 3 days thereafter. To evaluate the role of canonical WNT signaling for differentiation toward SAN-head lineage, XAV939 (5 μmol/L), was added from day 10 to day 17. To evaluate the role of TGFβ signaling for differentiation toward SAN-TZ lineage, TGFβ2 (R&D Systems, #302-B2-002/CF; 5 ng/mL) was added from day 4 to day 6.

## RT-qPCR

Total RNA of day 19 hiPSC-derived cultures was isolated using Nucleospin RNA kit (Machery Nagel, # MN740955.50) according to the manufacturer's instructions. Reverse transcription was performed using Superscript II (Thermo Fisher Scientific, #18064071) with oligo dT primers (125 μmol/L). qPCR was performed on the LightCycler 2.0 Real-Time PCR system (Roche Life Science). Primer pairs were designed to span an exon-exon junction or at least one intron (*Supplementary file 6*). qPCR mix was prepared using the LightCycler 480 SYBR Green I Master (Roche, #4887352001), primers (1 μmol/L), and cDNA (equivalent to 10 ng RNA). Amplification of target sequences was performed using the following protocol: 5 min at 95°C followed by 45 cycles of 10 s at 95°C, 20 s at 60°C, and 20 s at 72°C. Data analysis was performed using LinRegPCR program (*Ruijter et al., 2009*). For data normalization, two experimentally assessed reference genes, RPLP0 and GUSB, were used.

## Immunofluorescence staining

Cells cultured as a confluent monolayer on glass coverslips were fixed with 4% paraformaldehyde. Permeabilization was performed with 0.1% Triton-X (Sigma-Aldrich #T8787) and a blocking step was carried out with 4% swine serum (Jackson ImmunoResearch, #014-000-121) for 1 hr. Primary and secondary antibodies were diluted in 4% swine serum as stated in the key resource table and incubated at room temperature for 1 hr or at 4°C overnight. Cell nuclei were stained with DAPI (Sigma-Aldrich #D9542). Imaging was carried out with Leica TCS SP8 X DLS confocal microscope. Data visualization and processing was performed with the Leica LAS-X software.

## Single cell patch clamp

Day 16 cardiomyocytes were dissociated using ×1 TryPLE Select (Thermo Fisher Scientific #12563011) and plated at a density of 7.0^10$^3$ per coverslip. After 1 week, cells with a smooth surface and intact membrane were chosen for measurements. Action potentials were recorded at 37°C with the amphotericin-B-perforated patch clamp technique using a Axopatch 200B Clamp amplifier (Molecular Devices Corporation). Measurements were carried out in Tyrode's solution containing 140 mmol/L NaCl, 5.4 mmol/L KCl, 1.8 mmol/L $CaCl_2$, 1.0 mmol/L $MgCl_2$, 5.5 mmol/L glucose, and 5.0 mmol/L HEPES. pH was adjusted to 7.4 with NaOH. Pipettes (borosilicate glass; resistance 1.5–2.5 MΩ) were filled with a solution containing 125 mmol/L potassium gluconate, 20 mmol/L KCl, 10 mmol/L NaCl, 0.4 mmol/L amphotericin-B, and 10 mM HEPES, pH was adjusted to 7.2 with KOH. Signals were low-pass-filtered (cutoff frequency 10 kHz) and digitized at 40 kHz. Action potentials were corrected for the estimated change in liquid junction potential (*Barry and Lynch, 1991*). Data acquisition and analysis were performed using custom software.

## Immunohistochemistry on mouse heart tissue

Paraffin-embedded hearts were sectioned at 7 μm. Sections were mounted onto silane-coated slides, deparaffinized in xylene, rehydrated in graded ethanol series and washed in phosphate-buffered saline (PBS, pH 7.4). Heat-induced antigen retrieval was performed using unmasking solution (Vector

Labs #H-3300–250). Sections were incubated with primary antibodies (Key resources table) diluted in 4% bovine serum albumin (BSA; Sigma-Aldrich #A7906) at 4°C overnight. After washing in TBST buffer (25 mM Tris, 150 mM NaCl, 2.5 mM KCl, and 0.5% Tween w/v) sections were incubated with fluorochrome-conjugated secondary antibodies at room temperature for 2 hr in the dark. Sections were washed in TBST, stained with DAPI (Sigma-Aldrich #D9542) and mounted in PBS-glycerol (1:1). Imaging was performed with Leica DMI6000 inverted microscope.

## Flow cytometry

Day 18–20 cardiomyocytes were dissociated using ×1 TrypLE Select (Thermo Fisher Scientific #12563011). For intracellular staining, cells were fixed and stained using the FIX & PERM kit (Thermo Fisher Scientific; #GAS004) according to the manufacturer's instructions. For cell surface antigens, the antibody was added to the cell suspension resuspended in a buffer containing 10% BSA (Sigma-Aldrich, #A8022) and 0.5 M EDTA (Thermo Fisher Scientific #15575020). All antibody incubations were performed for 30 min on ice protected from light. Acquisition was performed on FacsCanto II Cell Analyzer (Beckton Dickinson). Data was analyzed using FlowJo version 10. Antibody information is provided in the Key resources table.

## Cell sorting for single cell RNA-seq

Single cell sequencing was performed using SORT-seq method (*Muraro et al., 2016*). Cells from one representative differentiation were collected at different stages (days 0, 4, 5, 6, and 10). At the end time point on day 19, cells from two independent differentiations were collected to ascertain reproducibility. For each time point, cells were sorted into two (D0–10, D19 SANCM$^{TGFB2}$) or three (D19 SANCM and D19 ACM) 384-well plates, each well containing an oil droplet with barcoded primers, spike-ins, and dNTPs. Preparation of single cell libraries was performed using the CEL-Seq2 protocol (*Muraro et al., 2016*; *Hashimshony et al., 2016*). Paired-end sequencing was performed on the NextSeq500 platform using 1×75 bp read length kit.

## Bioinformatic analysis

### Reference genome annotation

Mapping was performed using BWA-MEM against the (human) genome assembly GRCh38 (hg38). Count matrices were generated using MapAndGo, filtering reads with a minimum quality score of 60 and no alternative hits.

### scRNA-seq data pre-processing, normalization and batch correction, clustering, differential gene expression, cell-type identification and visualization

Data analysis was performed using the R toolkit Seurat versions 3 and 4 (*Stuart et al., 2019*, *Hao et al., 2021*). Data QC and pre-processing, dimensional reduction, clustering, and differential gene expression were performed according to the standard workflow (https://satijalab.org/seurat/). Briefly, high-quality single cells collected on D19 were selected according to the following parameters: gene count >1000 and <9000, mRNA molecule count <60,000 and mitochondrial gene count <50%. The filters for the time series dataset were set as per the following: gene count >600; mRNA molecule count <100,000; mitochondrial gene count <50%. Next, normalization, scaling, and identification of variable features (nfeatures = 3000) based on variance stabilizing transformation ('vst') was performed using the SCTransform command (*Hafemeister and Satija, 2019*). Since technical plate-to-plate variations were observed, SCTransform data integration was performed by normalizing each dataset individually, identifying integration anchors within the datasets collected on the same time point and integrating the datasets. Dimensionality reduction was performed using PC analysis (PCA) and UMAP with the top 15 PCs (day 19 datasets, *Figure 3* and *Figure 7*), top 20 PCs (day 0–19 SANCM dataset, *Figure 4*) and seed set to 2020. For cell clustering, a KNN (K-nearest neighbor) graph was constructed based on euclidean distance in PCA space and clusters were identified using the Louvain algorithm, as implemented in the FindNeighbors and FindClusters command. Identified clusters were then visualized in a UMAP using the DimPlot command. For differential expression testing and visualization, LogNormalization was performed according to the standard workflow on the uncorrected dataset and differential gene expression was determined using Wilcoxon rank sum test. Differentially expressed gene lists show genes, which are expressed in at least 25% in either of the two fractions of cells

and limited to genes, which are differentially expressed (on average) by at least 0.25-fold (log-scale) between the two compared cell fractions. Cell type-specific marker genes were used to annotate cell clusters. VlnPlot, FeaturePlot, and DoHeatMap commands were used to visualize gene expression.

### Pseudotime and trajectory inference

For the reconstruction of transcriptional trajectories from the mesodermal stage (day 4) to SANCM (day 19), the URD algorithm was used (*Farrell et al., 2018*). hiPSC clusters (D0_1, D0_2) were excluded as we reasoned that cell lineage diversification will not occur before mesoderm induction. A small endoderm-like cluster (D4_2) was also excluded. Identification of highly variable genes, PCA and tSNE projection (RunTSNE command, dims = 1:20) were performed using Seurat, as described above. The Seurat object was converted into a URD object. All steps were performed according to the manual provided by the Schier lab (https://schierlab.biozentrum.unibas.ch/urd).

Briefly,KNN graph was calculated using k=100 and poorly connected cells (outliers) were removed. Outliers were identified as cells, which are unusually far from their nearest neighbor and their 20th nearest neighbor (based on their distance to their nearest neighbor). Next, transition probabilities were calculated between transcriptomes to connect cells with similar gene expression patterns and a diffusion map was constructed using KNN = 50 and global sigma = 12. Diffusion map was visualized and assessed by plotting diffusion component pairs using PlotDimArray function. Then, the root of the specification tree was defined (cells in cluster D4_1, corresponding to mesoderm stage) and pseudotime was assigned to each cell by simulated 'floods' (n=100, minimum.cells.flooded=2), using previously calculated transition probabilities. The tips of the trajectory tree were assigned using clusters derived from terminally differentiated cells (day 19). Cluster 7 (*Figure 3A*) was not assigned as tip cluster as those cells appear to be halted during differentiation. Clusters 4, 5, 6, and 8 were used as tip clusters corresponding to SAN-TZ, SAN-tail, SAN-head, and proepicardial-like cells, respectively. Trajectories from the tips back to the root were identified using biased random walks with the following parameters: optimal.cells.forward=50, max.cells.back=80; n.per.tip=25,000,, root.visits= 1, max.steps=5000. In order to build the developmental trajectory and branching tree structure, the visitation frequency of each cell was determined by the random walks from each tip. Visitation frequencies were visualized to ensure a well-connected tree structure from the tips to the root. Lastly, the branching tree structure was constructed using the following parameters: divergence.method = 'preference', cells.per.pseudotime.bin=35, bins.per.pseudotime.window=10, save.all.breakpoint. info=T, p.thresh=0.000001. Gene expression within the dendrogram was visualized using the plotTree command. Differential gene expression between different segments of the developmental tree were performed using the markersAUCPR command (auc.factor=0.9, effect.size=0.4, frac.must.express=0. 5). Slingshot analysis was performed as described in the Bioconductor vignette.

## GO enrichment analysis

GO enrichment analysis was performed using Protein Analysis Through Evolutionary Relationships (PANTHER) Classification System version 16.0, release date 2020-12-01 (*Ashburner et al., 2000*; *Carbon et al., 2021*).

## Statistical analysis

Statistical analysis was carried out in GraphPad Prism version 9.1.0 for Windows GraphPad Software, San Diego, CA, https://www.graphpad.com/. Data were represented as mean ± s.e.m. (standard error of the mean). Non-parametric tests were performed in all cases. Number of samples (n) and the method used to test statistical significance are stated in each figure legend. $p < 0.05$ was considered statistically significant.

## Acknowledgements

We thank Berend Hooibrink from the Flow Cytometry facility, Department of Medical Biology, for assistance with cell sorting, Corrie de Gier-de Vries, Department of Medical Biology, for help with histology of mouse hearts and Likhitha Puliyadi, Department of Medical Biology for assistance with RT-qPCR. We also gratefully acknowledge funding from the European Research council starting grant 714866 and associated proof-of-concept grant 899422, Health Holland LentiPace II, Horizon 2020 Eurostars (E114245 and E115484), Dutch Research Council Open Technology Program 18485

to GJJB; Netherlands Organization for Health Research and Development (ZonMW), ZonMW TOP 40-00812-98-17061 to VMC, ZonMW and the Dutch Heart foundation MKMD grant 114021512 and Dutch Heart Foundation Dekker fellowship 2020T023 to HDD.

## Additional information

### Competing interests
Gerard JJ Boink: reports ownership interest in PacingCure B.V. The other authors declare that no competing interests exist.

### Funding

| Funder | Grant reference number | Author |
|---|---|---|
| ZonMw | 114021512 | Harsha D Devalla |
| Hartstichting | 2020T023 | Harsha D Devalla |
| European Research Council | 714866 | Gerard JJ Boink |
| European Research Council | 899422 | Gerard JJ Boink |
| Dutch Research Council Open Technology Program | 18485 | Gerard JJ Boink |
| Horizon 2020 Eurostars | E114245 | Gerard JJ Boink |
| Horizon 2020 Eurostars | E115484 | Gerard JJ Boink |

The funders had no role in study design, data collection and interpretation, or the decision to submit the work for publication.

### Author contributions
Alexandra Wiesinger, Data curation, Software, Formal analysis, Validation, Investigation, Visualization, Writing – original draft; Jiuru Li, Formal analysis, Investigation, Visualization; Lianne Fokkert, Priscilla Bakker, Investigation, Visualization; Arie O Verkerk, Formal analysis, Supervision, Investigation, Visualization; Vincent M Christoffels, Project administration, Writing – review and editing; Gerard JJ Boink, Funding acquisition, Project administration, Writing – review and editing; Harsha D Devalla, Conceptualization, Supervision, Funding acquisition, Validation, Investigation, Methodology, Writing – original draft, Project administration, Writing – review and editing

### Author ORCIDs
Alexandra Wiesinger http://orcid.org/0000-0001-9078-6235
Arie O Verkerk http://orcid.org/0000-0003-2140-834X
Vincent M Christoffels http://orcid.org/0000-0003-4131-2636
Harsha D Devalla http://orcid.org/0000-0001-5343-1021

### Decision letter and Author response
Decision letter https://doi.org/10.7554/eLife.76781.sa1
Author response https://doi.org/10.7554/eLife.76781.sa2

## Additional files

### Supplementary files
• Transparent reporting form

• Supplementary file 1. Differentially expressed genes in sinoatrial node-like cardiomyocyte (SANCM) and ventricular-like cardiomyocyte (VCM) clusters identified at day 19. Related to *Figure 3* and *Figure 3—figure supplement 1*.

• Supplementary file 2. Differentially expressed genes in clusters identified in time series analysis.

Related to *Figure 4* and *Figure 4—figure supplement 1*.

• Supplementary file 3. Gene ontology (GO) term analysis of differentially expressed genes between common progenitor segment (1) and proepicardial segment (2) or myocardial segment (3). Related to *Figure 5*.

• Supplementary file 4. Gene ontology (GO) term analysis of differentially expressed genes between myocardial segment (3) and sinoatrial node (SAN)-TZ segment (4) or SAN-tail segment (5), or SAN-head segment (6). Related to *Figure 6*.

• Supplementary file 5. Differentially expressed genes in clusters identified on day 19 comparing sinoatrial node-like cardiomyocyte (SANCM), SANCM^TGFB2, and ACM. Related to *Figure 7*.

• Supplementary file 6. Sequences of primers used in the manuscript. Related to *Figure 1*, *Figure 2*, *Figure 5*, *Figure 6*.

### Data availability

Source data have been provided for electrophysiology data presented in figure 2 and figure 7. Single cell RNA sequencing data presented in figures 3, 4, 5, 6 and 7 is deposited in NCBI GEO repository under the accession number GSE189782. We have also provided differentially expressed gene lists of all clusters described in figure 3, 4, 5, 6 and 7 as supplementary files 1 - 5. Supplementary files 3 and 4 also contain gene ontology enrichment analyses related to figure 5 and 6. R scripts used for analysis of single cell RNA sequencing data are available on Github (https://github.com/wiesingera/transcriptional_roadmap_hiPSC-SANCM, copy archived at swh:1:rev:54c1c4329e96f8d7e2cedf0f740f761c243d2e4a).

The following dataset was generated:

| Author(s) | Year | Dataset title | Dataset URL | Database and Identifier |
|---|---|---|---|---|
| Alexandra W, Harsha DD | 2021 | A single cell transcriptional roadmap for human pacemaker cell differentiation | https://www.ncbi.nlm.nih.gov/geo/query/acc.cgi?&acc=GSE189782 | NCBI Gene Expression Omnibus, GSE189782 |

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
