## [Editor Report]

The manuscript by Wiesinger et al., demonstrates the differentiation of human induced pluripotent stem cells (iPSCs) into pacemaker cardiomyocytes. Authors have shown impressive analyses of sinoatrial node cardiomyocytes (SANCM) using scRNA-seq followed by a computational method namely Trajectory Inference (TI) to understand the diversification of SAN subtypes. The study further shows a key role for Wnt signaling in the critical branching of pacemaker cardiomyocytes and/or pro-epicardial cells. The authors also demonstrate that active TGFβ signaling promotes differentiation towards SAN transitional cells.

---

## [Decision Letter]

**Decision letter after peer review:**

Thank you for submitting your article "A single cell transcriptional roadmap of human pacemaker cell differentiation" for consideration by *eLife*. Your article has been reviewed by 2 peer reviewers, and the evaluation has been overseen by a Reviewing Editor and a Senior Editor. The reviewers have opted to remain anonymous.

The reviewers have discussed their reviews with one another, and the Reviewing Editor has drafted this letter to help you prepare a revised submission.

Essential revisions:

Both expert reviewers are in general agreement that this manuscript adds to our knowledge base of how pacemaker cells differentiate. Revisions requested below include:

1) Confirmation of the pseudotime trajectory inference method;

2) Functional validation of specific markers found in this study;

3) Experiments exploring other signaling pathways that increase or decrease the efficiency in the creation of the different subtypes of SANCMs, or a more detailed evaluation of when the hiPSC based strategy begins to overlap with heart development and a characterization of the role of the newly identified genetics target(s) in SANCM subtypes differentiation in vivo.

*Reviewer #1 (Recommendations for the authors):*

Wiesinger et al., demonstrates the differentiation of human induced pluripotent stem cells (iPSCs) into pacemaker cardiomyocytes and study the time-course trajectory of the subtype specification highlighting the role of WNT/TGF β signaling.

The study can be a significant tool to the field of cardiac regeneration. Manuscript reads well with clear and detailed omics outline. Having said that the study seems to miss conceptual novelty. Sino atrial node cardiomyocyte differentiation (SAN-CM) from iPS cells is a known information. Authors also discuss the WNT/BMP/TGF β role in pacemaker cell formation along with the respective public information available on the markers of subtypes (SAN-head, SAN-tail and SAN-TZ). However, the crucial contribution here is the time course trajectory inference and identification of the role of WNT/TGF β in pushing the fate of different SAN-CM subtypes. Thus the results do support the conclusions overall despite the above caveat. Some of the questions/concerns that arose while assessing the manuscript is as follows:

1. Pseudotime trajectory inference using URD is a very good tool. Have the authors confirmed this by any other TI method? Just to confirm that the prediction is not biased due to the setting or the particular algorithm in a TI method.

2. Figure 1C shows a significantly higher TNNT2 in SAN-CMs than VCM. Does this owe to an increase in the maturation status of SAN-CM overall? This shouldn't be the case as authors show that UMAP from Figure 3 Cluster 2 have a high HOPX level indicative of a mature CM in ventricular myocytes. Especially in the context of differential NKX2.5 expression in Figure 1C, a clarity to this observation will strengthen the data.

3. SAN-CM scRNA-seq analysis by Liang et al., (2021) shows the marker VSNL1 as key factor in mouse SAN cell specification. Here the authors knock-down this marker and find that the beating ability of cells slow down/decrease. It is interesting that in this study authors do find VSNL1 as a SAN -specific marker in human SAN-CMs. Functional validation of specific/enriched markers found in this study can provide key information on the absolute requirement or redundancy of a specific subtypes in disease conditions. Such experiments may significantly strengthen the data.

*Reviewer #2 (Public Review):*

In the manuscript titled "A single cell transcriptional roadmap of human pacemaker cell differentiation," the authors seek to delineate the cell fate decisions that occur during the in vitro differentiation of human pacemaker cells (SANCM) from hiPSCs. The authors first compare marker expression and functional properties of differentiated SANCM and VCM cells, and establish that the SANCM cells have the expected characteristics of pacemaker cells. Single cell RNA sequencing was then used to explore the heterogeneity of the differentiated cells and illustrate the separate clustering of VCM and SANCM cells. The scRNAseq data was used to identify and characterize the different SANCM subtypes generated by the differentiation process. scRNAseq was then used to analyze samples from different stages of reprogramming and highlighted the changes in the transcriptome during the differentiation process. In addition, pseudotime analysis was performed in conjugation with pharmacological manipulation to show how WNT and TGF-beta signaling affect the stepwise progression of hiPSCs into the identified different SANCM subtypes. This study provides evidence for the presence of different SANCM subtypes generated by the SANCM differentiation process as well as illustrates the role of the WNT and TGF-beta in generating these different clusters of SANCM cells. Additional validation of the SANCM heterogeneity during the in vitro differentiation process as well as additional evidence of novel mediators of the acquisition of the unique SANCM subtype identity would strengthen the impact of this manuscript.

Specific suggestions:

1. The first scRNAseq experiment highlights the transcriptional differences between VCM and SANCM clusters, however, these differences are to be expected. This data also supports the hypothesis that the SANCM differentiation leads to a heterogeneous population. Additional bioinformatic analyses into the differences between these different clusters may provide more novel insights and could provide molecular targets to explore in vivo during embryonic development. For example, the identification of Vsnl1 and Gnao1 are promising gene candidates that should be further explored during multiple timepoints of heart development and validated with quantification. This data would provide complementary evidence that this differentiation process recapitulates what happens in vivo. Immunofluorescent staining of select markers of different scRNAseq clusters should also be provided to confirm the identified cluster-specific differentially expressed genes.

2. The final portion of the manuscript further establishes the specific roles of the WNT and TGF-beta components of the differentiation protocol, but requires additional experiments to show that the heterogeneity is affected at the single cell level when these pathways are altered (such as immunofluorescence staining to show that fewer cells are expressing that gene of interest rather than a systemic change seen by qPCR). Being that the significant roles of WNT and TGF-beta are to be expected due to the presence of chemical modulators of those pathways are present in the differentiation protocol, this manuscript would benefit from experiments exploring other signaling pathways that increase or decrease the efficiency in the creation of the different subtypes of SANCMs, or a more detailed evaluation of when the hiPSC based strategy begins to overlap with heart development and a characterization of the role of the newly identified genetics target(s) in SANCM subtypes differentiation in vivo.

---

## [Author Response]

Reviewer #1 (Recommendations for the authors):Wiesinger et al., demonstrates the differentiation of human induced pluripotent stem cells (iPSCs) into pacemaker cardiomyocytes and study the time-course trajectory of the subtype specification highlighting the role of WNT/TGF β signaling.The study can be a significant tool to the field of cardiac regeneration. Manuscript reads well with clear and detailed omics outline. Having said that the study seems to miss conceptual novelty. Sino atrial node cardiomyocyte differentiation (SAN-CM) from iPS cells is a known information. Authors also discuss the WNT/BMP/TGF β role in pacemaker cell formation along with the respective public information available on the markers of subtypes (SAN-head, SAN-tail and SAN-TZ). However, the crucial contribution here is the time course trajectory inference and identification of the role of WNT/TGF β in pushing the fate of different SAN-CM subtypes. Thus the results do support the conclusions overall despite the above caveat. Some of the questions/concerns that arose while assessing the manuscript is as follows:1. Pseudotime trajectory inference using URD is a very good tool. Have the authors confirmed this by any other TI method? Just to confirm that the prediction is not biased due to the setting or the particular algorithm in a TI method.

We thank the reviewer for this suggestion. To cross check findings from URD, we analyzed our data using an additional trajectory inference method, Slingshot (Street et al., 2018), (Revised Figure 5 —figure supplement 1 A-C), which has been shown to be robust in ordering the cells and assigning them in appropriate trajectory branches (Saelens et al., 2019). Similar to URD, Slingshot identified the correct order of the cells in our dataset. Furthermore, this approach also yielded a similar number of trajectories. However, one difference was that Slingshot did not recognize day 5 as part of the main trajectory and instead connected cells from day 4 directly to day 6, with day 5 cells forming an individual branch/lineage (Figure 5 —figure supplement 1B). We hypothesize that this outcome reflects the transcriptional burst induced by the treatment with pacemaker differentiation factors (BMP4, retinoic acid and others) introduced in the culture on day 4. Nevertheless, similar to URD, Slingshot identified a lineage branching off towards the proepicardial cells around day 10, and two separate cardiomyocyte lineages. These results are consistent with the clustering outcome shown in Figure 4B, where cells collected on day 19 of differentiation form three individual groups (SAN-head/SAN-tail, SAN-TZ and proepicaridal cells). The reason we opted for URD in our study is that it allows customized assignment of terminally differentiated endpoints through integration of secondary clustering results. We utilized this feature to annotate all three pacemaker subpopulations, i.e., SAN-head, SAN-tail and SAN-TZ as well as the proepicardial lineage identified through secondary clustering analysis in Figure 3, resulting in trajectory assignment for these four different lineages, instead of three. All in all, predictions made by URD and Slingshot are very comparable.

2. Figure 1C shows a significantly higher TNNT2 in SAN-CMs than VCM. Does this owe to an increase in the maturation status of SAN-CM overall? This shouldn't be the case as authors show that UMAP from Figure 3 Cluster 2 have a high HOPX level indicative of a mature CM in ventricular myocytes. Especially in the context of differential NKX2.5 expression in Figure 1C, a clarity to this observation will strengthen the data.

The reviewer is right that SANCM express higher *TNNT2* mRNA. However, we did not observe any differences between the subtypes in TNNT2 protein expression by flow cytometry (Figure 1B). Moreover, *ACTN2* expression was similar between the two groups (Figure 1C) and thus it is unlikely that there is a difference in their maturation states. We wonder whether this could be due to more efficient translation of *TNNT2* in VCM, possibly reflecting their contractile function.

3. SAN-CM scRNA-seq analysis by Liang etal (2021) shows the marker VSNL1 as key factor in mouse SAN cell specification. Here the authors knock-down this marker and find that the beating ability of cells slow down/decrease. It is interesting that in this study authors do find VSNL1 as a SAN -specific marker in human SAN-CMs. Functional validation of specific/enriched markers found in this study can provide key information on the absolute requirement or redundancy of a specific subtypes in disease conditions. Such experiments may significantly strengthen the data.

We agree with the reviewer that functional validation of genes enriched in pacemaker subpopulations would be valuable. However, it is beyond the scope of this study as a thorough analysis would require knockdown of at least a few genes per subpopulation. Moreover, to be able to evaluate the requirement/redundancy of subpopulations in disease (for e.g., transitional cells), we should ideally test the functional effect of knockdown in engineered models that incorporate both pacemaker and atrial cells. We are currently developing such models and expect to be able to gain more insights on these aspects in the future.

Reviewer #2 (Public Review):1. The first scRNAseq experiment highlights the transcriptional differences between VCM and SANCM clusters, however, these differences are to be expected. This data also supports the hypothesis that the SANCM differentiation leads to a heterogeneous population. Additional bioinformatic analyses into the differences between these different clusters may provide more novel insights and could provide molecular targets to explore in vivo during embryonic development. For example, the identification of Vsnl1 and Gnao1 are promising gene candidates that should be further explored during multiple timepoints of heart development and validated with quantification. This data would provide complementary evidence that this differentiation process recapitulates what happens in vivo. Immunofluorescent staining of select markers of different scRNAseq clusters should also be provided to confirm the identified cluster-specific differentially expressed genes.2. The final portion of the manuscript further establishes the specific roles of the WNT and TGF-beta components of the differentiation protocol, but requires additional experiments to show that the heterogeneity is affected at the single cell level when these pathways are altered (such as immunofluorescence staining to show that fewer cells are expressing that gene of interest rather than a systemic change seen by qPCR). Being that the significant roles of WNT and TGF-beta are to be expected due to the presence of chemical modulators of those pathways are present in the differentiation protocol, this manuscript would benefit from experiments exploring other signaling pathways that increase or decrease the efficiency in the creation of the different subtypes of SANCMs, or a more detailed evaluation of when the hiPSC based strategy begins to overlap with heart development and a characterization of the role of the newly identified genetics target(s) in SANCM subtypes differentiation in vivo.

We thank the reviewer for their suggestions. In response to comment 1, bioinformatic analysis presented in figure 3 as well as accompanying supplement files provide detailed insights into the transcriptional differences between the various SAN subpopulations and additional analysis will not add new information. We agree with the other suggestions provided by the reviewer and are currently working on obtaining additional data to support our conclusions.